# Cancer Risk Assessment and Geochemical Features of Granitoids at Nikeiba, Southeastern Desert, Egypt

Ahmed E. Abdel Gawad [1],*, Hassan Eliwa [2], Khaled G. Ali [1], Khalid Alsafi [3], Mamoru Murata [4], Masoud S. Salah [1] and Mohamed Y. Hanfi [1,5],*

1 Nuclear Materials Authority, P.O. Box 530, Maadi, Cairo, Egypt; khaled_ali@yahoo.com (K.G.A.); masoudsalah85@gmail.com (M.S.S.)
2 Geology Department, Faculty of Science, Minufiya University, P.O. Box 32511, Shebin El Kom, Egypt; eliwa98@yahoo.com
3 Department of Radiology, Consultant Medical Physics, King Abdulaziz University Hospital (KAUH), P.O. Box 80215, Jeddah, Saudi Arabia; kalsafi@kau.edu.sa
4 Department of Geosciences, Faculty of Science Naruto, Naruto University of Education, National University of Corporation, Tokushima 772-8502, Japan; atarumm@naruto-u.ac.jp
5 Institute of Physics and Technology, Ural Federal University, St. Mira, 19, 620002 Ekaterinburg, Russia
* Correspondence: gawadnma@gmail.com (A.E.A.G.); mokhamed.khanfi@urfu.ru (M.Y.H.)

**Abstract:** Different rock types (syenogranite, alkali feldspar granite and quartz syenite intruded by microgranite dikes and quartz veins) were investigated in the Nikeiba region in Egypt. The main components of the studied intrusive rocks, comprised of granites and quartz syenite, are plagioclase, amphibole, biotite, quartz and K-feldspar in different proportions. Ground gamma ray measurements show that syenogranite, quartz syenite and microgranite dikes have the highest radioactivity (K, eU, eTh and their ratios) in comparison with alkali feldspar granite. Geochemically, syenogranite, alkali feldspar granite and quartz syenite are enriched with large-ion lithophile elements (LILE; Ba, Rb, Sr) and high field-strength elements (HFSE; Y, Zr and Nb), but have decreased Ce, reflecting their alkaline affinity. These rocks reveal calc–alkaline affinity, metaluminous characteristics, A-type granites and post-collision geochemical signatures, which indicates emplacement in within-plate environments under an extensional regime. U and Th are increased in syenogranite and quartz syenite, whereas alkali feldspar granite shows a marked decrease in U and Th. The highest average values of $A_U$ ($131 \pm 49$ Bq·kg$^{-1}$), $A_{Th}$ ($164 \pm 35$) and $A_K$ ($1402 \pm 239$) in the syenogranite samples are higher than the recommended worldwide average. The radioactivity levels found in the samples are the result of the alteration of radioactive carrying minerals found inside granite faults. The public's radioactive risk from the radionuclides found in the investigated granitoid samples is estimated by calculating radiological risks. The excess lifetime cancer (ELCR) values exceed the permissible limit. Therefore, the granitoids are unsuitable for use as infrastructure materials.

**Keywords:** Nikeiba; gamma ray measurements; geochemistry; granite; radiological risk

## 1. Introduction

It has been recognized that felsic igneous rocks contain higher uranium concentrations than other rock types [1,2]. The common spatial association of granite and acidic volcanics suggests that such rocks are potential sources of uranium [1,3–8]. In the last decades, comprehensive programs for uranium exploration in the post-orogenic granitoids in the Eastern Desert have been carried out. These programs led to the discovery of some U-mineralization related to the younger granites (550–590 Ma) [9]. Most of this mineralization is restricted to granites such as Ras Abda [10,11], El-Erediya [12,13], El Missikat [14] and Gattar [15–18] in the Central Eastern Desert. In 1975, an important disseminated-type U-mineralization was discovered in the Um Ara-Um Shilman and El Sela younger granite plutons in the Southeastern Desert [19–28].

The public's focus of human exposure to ionizing radiation has heightened. After all, natural-source radioactivity is responsible for the vast majority of human exposure to radiation [29]. According to the geology of each place, natural uranium and thorium can be found in various amounts in all terrestrial materials [30].

In recent years, there has been a lot of discussion about the radioactive dangers of exposure to construction supplies [31]. The background level of radiation in the environment is influenced by terrestrial radionuclides and their offspring. Mineral, geochemical and physicochemical variables all play a role in their surroundings [31,32].

In this work, we provide a statistically significant database on radioelement contents in granitoids to show that in situ radioelement measurements are valuable datasets for evaluating different variations in granitic terrains of Nikeiba. Further, the present study aims to identify the geochemical features of granites and quartz syenite in the study area. The tectonic environment as well as potential radioactive elements of the rocks were clarified. We also aim to detect the radioactive concentrations in the examined granitoids, which might potentially be used in infrastructure. Furthermore, several radioactive variables were found in the assessment of public exposure to gamma radiation via the estimation of radiological dangers.

## 2. Geologic Setting

The exposed rock units are represented by metavolcanics, syenogranite, alkali feldspar granite and quartz syenite crosscut by microgranite dikes as well as quartz veins at Nikeiba [33–35]. Metavolcanics form a thick sequence of stratified lava flows interbanded with their pyroclastics and intruded by granitoids (syenogranite, alkali feldspar granite and quartz syenite) (Figure 1).

Syenogranite is medium- to coarse-grained, whitish to pale pink, buff or reddish brown in color, jointed, strongly weathered and exfoliated. It contains xenoliths up to 1 m of subangular metatuffs along their outer periphery. It is composed mainly of K-feldspar, quartz, plagioclase and biotite. Zircon, allanite, titanite, apatite, fluorite and iron oxides are the main accessories, while chlorite and epidote are the main alteration products. Alkali feldspar granite is coarse-grained, whitish or yellowish to pale white in color, highly weathered and holocrystalline equigranular rock. Porphyritic and micrographic textures are observed. It is essentially composed of K-feldspar, quartz, plagioclase and biotite. Zircon, apatite and iron opaque minerals are accessories, whereas chlorite and seicite are alteration products. Quartz syenite is medium- to coarse-grained, dark grey to pale greenish grey or pale pink in color and moderate to high relief. It is highly weathered, exfoliated, holocrystalline, hypidiomorphic granular rock and microscopically composed of K-feldspars, quartz, biotite, riebeckite, arfvedsonite and a very subordinate amount of plagioclase. Zircon, apatite, allanite, iron oxides and opaque minerals are accessories, whereas muscovite, chlorite, sericite, epidote, and carbonates are the main alteration products.

Pegmatites are very coarse-grained, buff to reddish in color and composed of K-feldspar, quartz, plagioclase, biotite and muscovite. They are found as an irregular body in syenogranite, with microgranite dikes and quartz veins dissecting granitoid rocks.

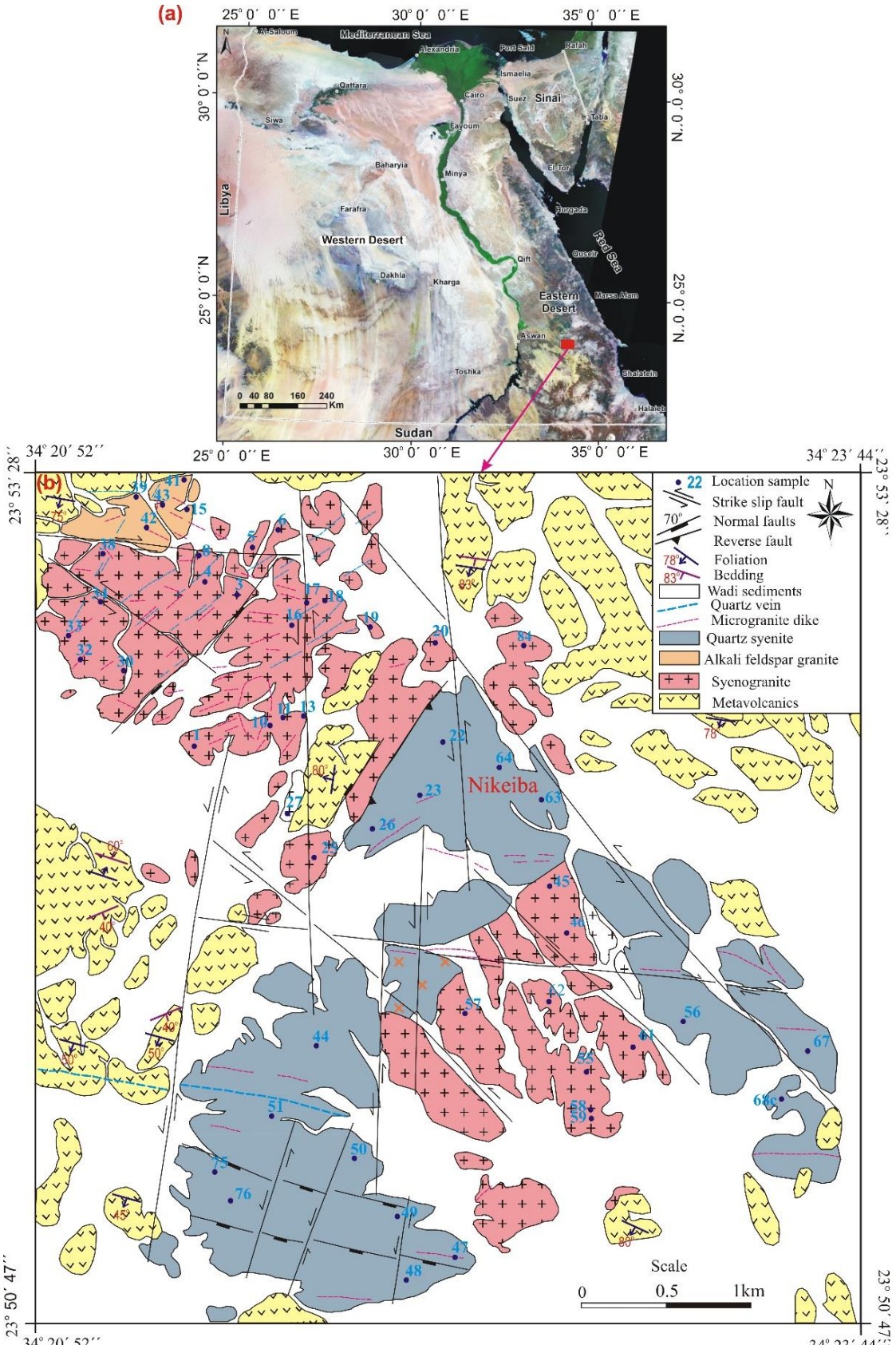

**Figure 1.** (**a**) Physiographic map and (**b**) detailed geologic map showing locations of the analyzed samples from Nikeiba, Southeastern Desert, Egypt.

## 3. Material and Analytical Methods

The ground gamma ray spectrometric measurements were conducted by using a Geophysica Brno GS-256 spectrometer. The instrument is manufactured by radiation solution Inc in Sugar City, ID, USA. having a 0.35 L sodium iodide (NaI) thallium activated detector. A measuring interval of 120 s was maintained in order to allow sufficient time to establish a stable spectrum. The ground spectrometry survey sites were selected to cover granitoid exposures in the study area. The radioelement measurements were taken using a single measurement of each site for equivalent thorium, eTh (ppm), equivalent uranium, eU (ppm), and potassium percent, K (%). The GS integrates a horizontal area of about 1 m diameter with nearly 25 cm of depth when in direct contact with the granitoid outcrops. The spectrometer is well-calibrated on artificial concrete pads at the Nuclear Materials Authority of Egypt before field survey. The pads contain known concentrations of potassium, uranium and thorium as described in Grasty et al. [36].

According to Clarke et al. [37], the eU/eTh ratio equals about 0.33 in granitic rocks. This ratio depends mainly on the mobile element (uranium), so the eU/eTh ratio is important for uranium exploration because it determines uranium-enriched areas. In order to get an idea about the remobilization of uranium in the area, the expected original contents of uranium are calculated by dividing eTh content by the Clarke value of the eTh/eU ratio (3–4) in granite according to Clarke et al. [37]. This is very helpful in defining the trends of uranium migration. Further, eU-eTh/3.5 enables the delineation of the limits between the negative values (leaching) and positive values (deposition).

Fifty-one samples were crushed and then powdered using agate mortar to avoid contamination. These sample powders were analyzed for major oxides and some trace elements at the Department of Geoscience, Shimane University, Matsue-City, Shimane Prefecture 690-8504, Japan. The analyses of the major and trace elements were carried out on glass discs prepared by fusing a 1.8 gm of the dry powdered mixed with 3.6 gm of alkali flux ($LiBO_2$: $LiB_4O_7$ = 1:4) [38] using a Rigaku RIX 2000 X-ray fluorescence spectrometer manufactured in Rigaku Corporation, Takatsuki-City, Osaka Prefecture 569-1149, Japan. Analytical precision, as calculated from replicate analyses, is 0.5% for major oxides and varies from 2–5% for trace elements of >80 ppm, 2–10% for trace elements of 10 to 80 ppm and 5–20% for trace elements of <10 ppm.

## 4. Results and Discussion

### 4.1. Radioactivity

Syenogranite is characterized by considerable radioelement content, which ranges from 2.2 to 8.5% for K, from 1.3 to 22.2 ppm for eU and from 25.6 to 56.3 ppm for eTh. Moreover, it exhibits the ratio ranges (0.04 to 0.56) for eU/eTh, 3.82 to 17.09 for eTh/K and −10.04 to 10.89 for eU-(eTh/3.5). The calculated standard deviations represent a low to moderate dispersion of the data around the calculated average values (Table 1).

Alkali feldspar granite is characterized by low radioelement content, ranging from 1.3 to 3.7% for K, from 1.3 to 6.8 ppm for eU and from 7.4 to 24.5 ppm for eTh. The eU/eTh ratio ranges from 0.1 to 0.8, eTh/K ratio ranges from 2.5 to 10.5 and eU-(eTh/3.5) ranges from −3.8 to 5.7. The calculated standard deviations represent low dispersion of the data around the calculated mean values for K and eU/eTh, but they represent moderate dispersion of the data around the arithmetic mean values of eU, eTh, eU-(eTh/3.5) and the (eTh/K) ratio.

The radioelement contents in quartz syenite varies between 2.5 to 5.5% for K, 2.3 to 13.8 ppm for eU and 20.6 to 30 ppm for eTh. Meanwhile, eU/eTh ratio ranges from 0.1 to 0.49, eTh/K ratio ranges from 4.2 to 9.9 and eU-(eTh/3.5) ranges from −5.03 to 5.43. The calculated standard deviations represent low dispersion of the data around the mean value except for eTh and eU-(eTh/3.5), which have moderate values.

Microgranite dikes have the highest radioelement contents among all the studied different rocks. They include a wide range of radioactive constituents; 1.3 to 11.3% for K, 6.5 to 41.3 ppm for eU and 28.7 to 371.1 ppm for eTh. Moreover, they also exhibit wide ratio ranges: 3.7 to 185.6 for eTh/K, 0.01 to 0.6 for eU/eTh and −83.6 to 16.7 ppm for

eU-(eTh/3.5). Their standard deviations represent very high dispersion of the data around the calculated mean value. This means that this rock unit has high U mobilization, and its ratios are not normal and form anomalous zones.

**Table 1.** Summary of the statistics of the surface distribution of the three radioelement potassium percentages, equivalent uranium and equivalent thorium (K%, eU (ppm), eTh (ppm), respectively) and their ratios at Nikeiba, Southeastern Desert, Egypt.

| Radioelements/St. Par. | K% | eU (ppm) | eTh (ppm) | eU/eTh | eTh/K | eU-(eTh/3.5) (ppm) |
|---|---|---|---|---|---|---|
| Syenogranite | | | | | | |
| Min. | 2.20 | 1.30 | 25.60 | 0.04 | 3.82 | −10.04 |
| Average | 4.24 | 9.88 | 38.36 | 0.26 | 9.30 | −1.08 |
| Max. | 8.50 | 22.20 | 56.30 | 0.56 | 17.09 | 10.89 |
| S.D. | 0.83 | 3.28 | 6.48 | 0.08 | 2.04 | 3.05 |
| No. | 246 | | | | | |
| Alkali feldspar granite | | | | | | |
| Min. | 1.30 | 1.30 | 7.40 | 0.10 | 2.50 | −3.80 |
| Average | 2.90 | 4.70 | 17.80 | 0.30 | 6.20 | −0.30 |
| Max. | 3.70 | 6.80 | 24.50 | 0.80 | 10.50 | 5.70 |
| S.D. | 0.50 | 1.60 | 3.70 | 0.10 | 1.60 | 1.70 |
| No. | 82 | | | | | |
| Quartz syenite | | | | | | |
| Min. | 2.50 | 2.3 | 20.60 | 0.10 | 4.2 | −5.03 |
| Average | 4.00 | 6.67 | 25.95 | 0.26 | 6.61 | −0.75 |
| Max. | 5.50 | 13.8 | 30.00 | 0.49 | 9.9 | 5.43 |
| S.D. | 0.54 | 2.12 | 2.19 | 0.08 | 1.07 | 2.07 |
| No. | 182 | | | | | |
| Microgranite dikes | | | | | | |
| Min. | 1.30 | 6.50 | 28.70 | 0.001 | 3.70 | −83.60 |
| Average | 5.20 | 17.20 | 107.2 | 0.20 | 36.10 | −13.40 |
| Max. | 11.30 | 41.30 | 371.1 | 0.60 | 185.6 | 16.70 |
| S.D. | 3.10 | 8.50 | 88.20 | 0.10 | 43.40 | 22.60 |
| No. | 48 | | | | | |

St. par.—Statistical parameters: Min.—Minimum; Max.—Maximum; No.—number of readings; S.D.—standard deviation.

The binary diagrams of K, eU and eTh values (Figure 2) show that Th has different distribution ranges for different granitic rock types. Both Th and K are generally considered immobile elements out of the K-metasomatism [39]. Thus, their concentrations can be used as indicators for magmatic fractionation and late magmatic variations, while U and K contents can be used to indicate hydrothermal alteration processes.

The granite samples plotted on the eTh (ppm) versus K % binary diagram can be classified into three groups (Figure 2a); the first group shows low K values (1 to 3.7%) and a limited eTh range (7 to 24 ppm), which is associated with alkali feldspar granite. The eTh—K binary diagram shows a reverse relation between eTh and K% in alkali feldspar granite due to the increase of Na% over K% during the late albitization process. The second group is characterized by similar contents of eTh and K, and K/eTh ratios ranging between 300 and 460, and is associated with quartz syenite. The third group includes samples

with K/eTh ratios of about 300 and is characterized by contemporaneous increases of eTh (25–56.3 ppm eTh) and K (2.2–8.5%K).

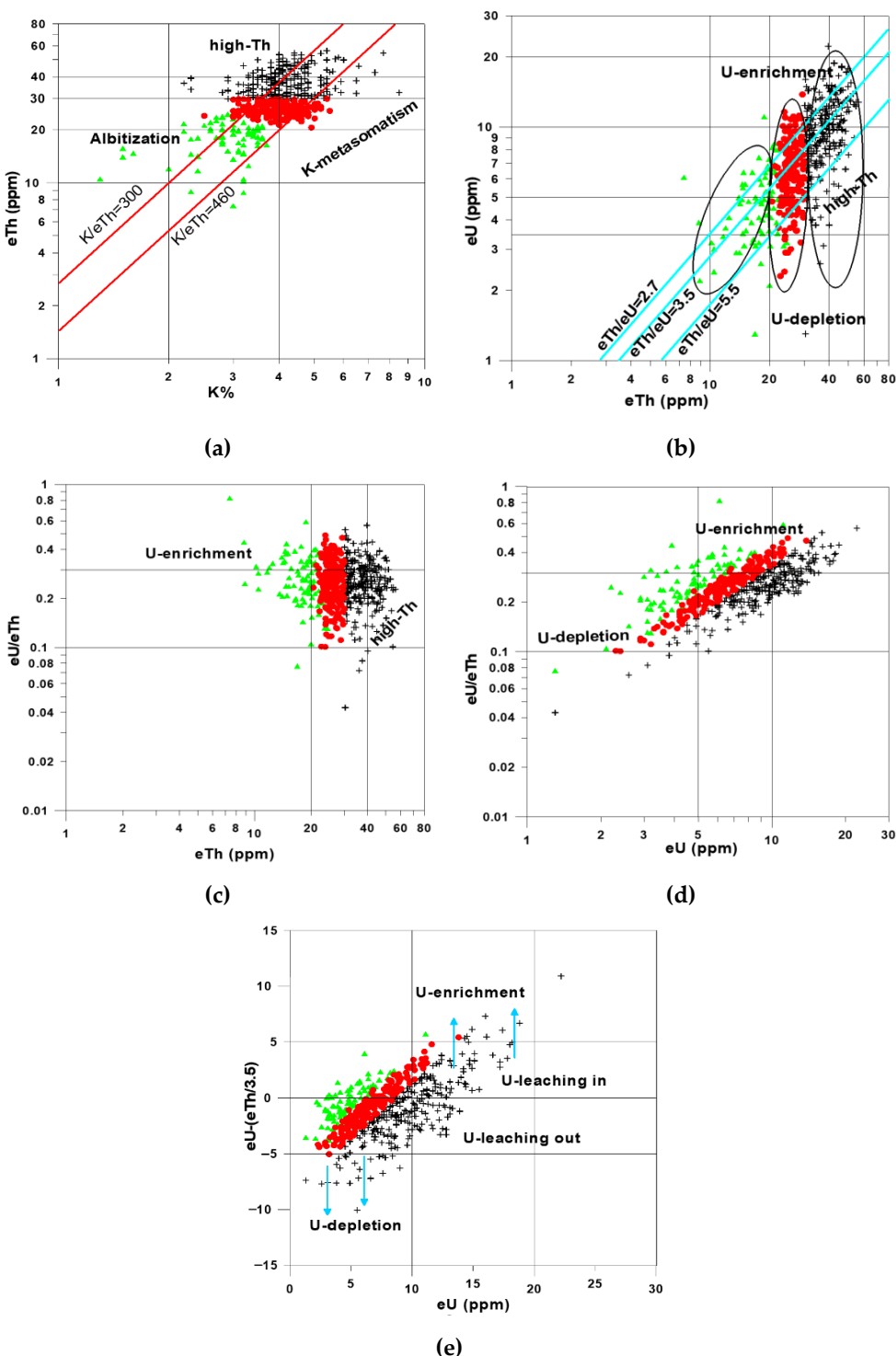

**Figure 2.** Radioactive element plots for ground gamma ray spectrometry measurements of syenogranite, alkali feldspar granite and quartz syenite at Nikeiba, Southeastern Desert, Egypt. Symbols are the same in the following diagrams (+ syenogranite, ▲ alkali feldspar granite and ● quartz syenite); diagram axes indicate data plots in logarithmic scale. (**a**) eTh (ppm) vs. K %; (**b**) eU (ppm) vs. eTh (ppm); (**c**) eU/eTh ratio vs. eTh (ppm); (**d**) eU/eTh ratio vs. eU (ppm); (**e**) eU-(eTh/3.5) vs. eU (ppm) binary diagrams.

The eU (ppm) versus eTh (ppm) binary diagram (Figure 2b) shows a linear relationship between eTh and eU, which means there is U depletion. Most of the samples are grouped around an eTh/eU ratio of 2.7 to 5.5 during fractionation. The eTh/eU ratio of some samples of alkali feldspar granite and albitized syenogranite also shows low values of U content. On the other hand, some episyenitized samples of syenogranite are enriched in U (17 to 25 eU ppm).

There is a reverse relationship between K and Th in alkali feldspar granite. Further, some samples of syenogranite show low K content with high Th, which may be related to albitization (Figure 2a). Limited K metasomatism with K values greater than 5% without high Th is recorded in syenogranite. On the other hand, some samples of syenogranite show high K content (>5%), which might have resulted from K metasomatism. Some samples with low U (<4 ppm eU) and high Th (>30 ppm eTh) are associated mainly with pegmatite pockets (Figure 2b). On the same Figure 2b, few samples with U and Th enrichment are encountered, which might be associated with episyenitized syenogranite along the fractures.

The eU/eTh ratio increases with eTh (Figure 2c) for most plotted alkali feldspar granite, syenogranite and quartz syenite samples. The eU/eTh ratio versus eU diagram (Figure 2d shows a strong direct slope, which indicates hydrothermal U enrichment.

Figure 2e shows that different samples of quartz syenite, syenogranite and alkali feldspar granite show higher uranium with positive values, indicate that the U content of these samples are leaching in. On the other hand, some samples have lower uranium with marked negative values, indicate that the U content of these samples is leaching out.

### 4.2. Geochemistry

Geochemical data were used to determine the characteristics of the studied uranium-fertile granites. Uranium fertility depends essentially on the composition of the host granites. Peraluminous, high K and low Na and Ca contents, and the two mica granites are considered as one of the main U fertile granite in the Eastern Desert of Egypt. Under certain conditions of alteration, high K calc–alkaline (HKCA) granites can form uranium ore with less tonnage. Moreover, uranium potentiality and processes responsible for uranium and thorium mineralization will be discussed.

The analyzed samples are indicated on the location map (Figure 1) and include syenogranite (32 samples), alkali feldspar granite (5 samples) and quartz syenite (14 samples).

Table 2 shows that $SiO_2$ wt% content increases from quartz syenite (65.08–71.12; avg. 68.02%), through syenogranite (70.61–76.57; avg. 73.72%) to alkali feldspar granite (77.18–78.68; avg. 78.01%). Harker variation diagrams reveal a general decrease in $TiO_2$, $Al_2O_3$, $Fe_2O_3$, MgO, MnO and CaO contents with increasing $SiO_2$, but a wide scattering of MgO in syenogranite is shown. Quartz syenite shows high alkalis content compared to alkali feldspar granite, which is $SiO_2$-rich.

Among the study rocks, quartz syenite contains the highest Ce, Nb, Zr, Ba and Sr contents, with averages of 237.6 ppm, 126.5 ppm, 554 ppm, 407 ppm and 100 ppm, respectively. Alkali feldspar granite contains lower contents of Sr, Zr, Nb and Ba than quartz syenite and syenogranite.

The distribution behavior of trace elements during magmatic differentiation can be traced by their variations with increasing $SiO_2$ (Figure 3). All samples are enriched in most large-ion lithophile elements (LILE; Ba, Rb, Sr) and high field-strength elements (HFSE; Y, Zr and Nb), but decrease in Ce, reflecting its alkaline affinity.

**Table 2.** Representative chemical (XRF) analyses of the Nikeiba granitoids, Southeastern Desert, Egypt. ($Fe_2O_3$ * total iron as ferric iron oxide.) For locations of sample sites refer to Figure 1.

| Rock | Syenogranite | | | | | | | | | | | | | | | | |
|---|---|---|---|---|---|---|---|---|---|---|---|---|---|---|---|---|
| S. No. | 1 | 3 | 4 | 5 | 6 | 8 | 10 | 11 | 13 | 16 | 17 | 18 | 19 | 20 | 25 | 27 | 30 |
| Major oxides (wt%) | | | | | | | | | | | | | | | | | |
| $SiO_2$ | 71.85 | 72.07 | 71.19 | 71.25 | 71.41 | 72.42 | 73.79 | 73.56 | 76.56 | 75.7 | 76.17 | 77.54 | 73.79 | 70.61 | 70.94 | 73.08 | 76.57 |
| $TiO_2$ | 0.19 | 0.15 | 0.18 | 0.18 | 0.19 | 0.13 | 0.14 | 0.12 | 0.06 | 0.07 | 0.07 | 0.07 | 0.14 | 0.19 | 0.18 | 0.17 | 0.08 |
| $Al_2O_3$ | 14.4 | 14.51 | 14.76 | 14.62 | 14.62 | 14.42 | 13.95 | 13.82 | 12.69 | 12.97 | 12.76 | 12.67 | 13.55 | 14.73 | 14.61 | 13.68 | 12.63 |
| $Fe_2O_3$ * | 2.74 | 2.40 | 2.71 | 2.77 | 2.84 | 2.26 | 1.95 | 1.97 | 1.50 | 1.50 | 1.39 | 0.43 | 2.16 | 2.95 | 2.83 | 2.53 | 1.37 |
| MnO | 0.06 | 0.05 | 0.06 | 0.06 | 0.06 | 0.06 | 0.04 | 0.04 | 0.02 | 0.04 | 0.03 | 0.01 | 0.04 | 0.07 | 0.09 | 0.05 | 0.02 |
| MgO | 0.15 | 0.08 | 0.12 | 0.12 | 0.13 | 0.07 | 0.13 | 0.08 | 0.02 | 0.03 | 0.03 | 0.02 | 0.12 | 0.12 | 0.04 | 0.12 | 0.00 |
| CaO | 0.80 | 0.66 | 0.90 | 1.06 | 0.91 | 0.63 | 0.63 | 0.74 | 0.33 | 0.55 | 0.56 | 0.49 | 0.73 | 0.85 | 0.71 | 0.83 | 0.29 |
| $Na_2O$ | 5.12 | 5.39 | 5.30 | 5.26 | 5.18 | 5.25 | 4.24 | 5.15 | 4.95 | 4.74 | 4.78 | 4.68 | 4.95 | 5.80 | 5.26 | 5.11 | 4.70 |
| $K_2O$ | 4.66 | 4.66 | 4.74 | 4.66 | 4.65 | 4.73 | 5.12 | 4.50 | 3.87 | 4.39 | 4.20 | 4.08 | 4.48 | 4.67 | 5.33 | 4.38 | 4.34 |
| $P_2O_5$ | 0.03 | 0.02 | 0.03 | 0.03 | 0.03 | 0.02 | 0.02 | 0.01 | 0.01 | 0.02 | 0.02 | 0.01 | 0.03 | 0.02 | 0.01 | 0.03 | 0.01 |
| Total | 100 | 99.99 | 99.99 | 100 | 100 | 99.99 | 100 | 99.99 | 100 | 99.99 | 99.99 | 99.99 | 99.99 | 100 | 100 | 99.98 | 100 |
| * CIPW norm | | | | | | | | | | | | | | | | | |
| Qz | 22.65 | 21.73 | 20.58 | 21.00 | 21.73 | 22.64 | 28.02 | 25.2 | 32.36 | 30.8 | 31.84 | 34.04 | 26.61 | 17.86 | 19.03 | 25.25 | 32.19 |
| Or | 27.54 | 27.54 | 28.01 | 27.54 | 27.48 | 27.95 | 30.29 | 26.59 | 22.87 | 25.97 | 24.85 | 24.14 | 26.47 | 27.6 | 31.5 | 25.88 | 25.65 |
| Ab | 43.32 | 45.61 | 44.85 | 44.51 | 43.83 | 44.42 | 35.84 | 43.58 | 41.89 | 40.06 | 40.4 | 39.56 | 41.89 | 49.08 | 44.51 | 43.24 | 39.77 |
| An | 2.55 | 1.63 | 2.48 | 2.52 | 2.91 | 1.81 | 3.13 | 1.30 | 0.98 | 1.11 | 0.92 | 1.48 | 1.52 | 0.36 | 0.51 | 1.45 | 0.55 |
| Cor | 0 | 0 | 0 | 0 | 0 | 0 | 0.27 | 0 | 0 | 0 | 0 | 0 | 0 | 0 | 0 | 0 | 0 |
| Di/en | 0.62 | 0.43 | 0.64 | 0.64 | 0.70 | 0.38 | 0.00 | 0.43 | 0.11 | 0 | 0 | 0 | 0.64 | 0.64 | 0.21 | 0.64 | 0 |
| Hy | 0.08 | 0 | 0 | 0 | 0 | 0 | 0 | 0 | 0 | 0 | 0 | 0 | 0 | 0 | 0 | 0 | 0 |
| Di/wo | 0 | 0.26 | 0.24 | 0.55 | 0.04 | 0.2 | 0 | 0.62 | 0.16 | 0.09 | 0.09 | 0.06 | 0.31 | 1.05 | 1.00 | 0.52 | 0.29 |

**Table 2.** *Cont.*

| | | | | | | | | | | | | | | | | | |
|---|---|---|---|---|---|---|---|---|---|---|---|---|---|---|---|---|---|
| Ac | 0 | 0 | 0 | 0 | 0 | 0 | 0 | 0 | 0 | 0.08 | 0.08 | 0.05 | 0 | 0 | 0 | 0 | 0 |
| Ilm | 0.13 | 0.11 | 0.13 | 0.13 | 0.13 | 0.13 | 0 | 0.09 | 0.04 | 0 | 0 | 0 | 0.09 | 0.15 | 0.19 | 0.11 | 0.04 |
| Hm | 2.74 | 2.40 | 2.71 | 2.77 | 2.84 | 2.26 | 0.33 | 1.97 | 1.50 | 0 | 0 | 0 | 2.16 | 2.95 | 2.83 | 2.53 | 1.37 |
| Ap | 0.07 | 0.05 | 0.07 | 0.07 | 0.07 | 0.05 | 0.13 | 0.02 | 0 | 0.13 | 0.10 | 0.03 | 0.07 | 0.05 | 0.02 | 0.07 | 0 |
| Tn | 0.30 | 0.23 | 0.28 | 0.28 | 0.30 | 0.15 | 1.86 | 0.18 | 0.09 | 1.41 | 1.32 | 0.41 | 0.23 | 0.27 | 0.19 | 0.28 | 0.14 |
| Ru | 0 | 0 | 0 | 0 | 0 | 0 | 0 | 0 | 0 | 0 | 0 | 0 | 0 | 0 | 0 | 0 | 0 |
| Elements (ppm) | | | | | | | | | | | | | | | | | |
| Ce | 155.80 | 175.00 | 135.30 | 143.70 | 155.20 | 146.80 | 128.20 | 110.90 | 27.90 | 76.40 | 56.70 | 51.90 | 105.90 | 138.50 | 252.40 | 128.30 | 41.20 |
| Th | 19.44 | 16.12 | 18.98 | 22.36 | 20.53 | 19.68 | 30.32 | 25.09 | 20.85 | 34.79 | 32.45 | 37.57 | 31.22 | 28.49 | 24.13 | 25.84 | 24.72 |
| U | 5.40 | 5.10 | | 4.60 | | | 7.60 | 4.20 | | 7.10 | | | | | | | 6.90 |
| Cr | 86.00 | 50.30 | 95.40 | 129.5 | 96.40 | 120.20 | 153.0 | 75.20 | 168.2 | 139.2 | 71.30 | 104.9 | 115.0 | 107.6 | 69.2 | 84.60 | 139.4 |
| Y | 42.70 | 39.80 | 45.00 | 46.10 | 45.40 | 47.10 | 56.40 | 46.80 | 36.80 | 55.00 | 49.10 | 45.70 | 52.10 | 51.40 | 54.90 | 49.00 | 48.40 |
| Nb | 96.92 | 76.18 | 85.99 | 87.20 | 85.36 | 90.85 | 141.1 | 102.5 | 110.5 | 113.3 | 102.6 | 104.5 | 125.6 | 130.8 | 88.37 | 116.5 | 151.5 |
| Zr | 360.7 | 365.00 | 378.6 | 394.0 | 411.8 | 351.6 | 371.7 | 310.0 | 142.5 | 206.4 | 169. | 207.2 | 313.5 | 511.0 | 562.1 | 342.3 | 166.8 |
| Ni | 0.30 | 2.00 | 0.70 | 1.20 | 0.30 | 0.60 | 3.20 | 1.30 | 1.50 | 0.60 | 0.20 | 0.70 | 0.20 | 0.30 | 0.70 | 0.50 | 0.10 |
| Ba | 338.7 | 414.5 | 467.0 | 435.1 | 450.2 | 382.8 | 260.3 | 285.3 | 66.10 | 88.50 | 79.50 | 86.60 | 200.3 | 297.0 | 156.0 | 237.2 | 74.40 |
| Pb | 9.80 | 7.80 | 9.70 | 9.40 | 8.70 | 7.00 | 10.50 | 9.10 | 7.30 | 8.20 | 6.70 | 6.70 | 8.10 | 11.40 | 10.70 | 8.70 | 5.00 |
| Rb | 147.4 | 149.5 | 151.3 | 151.8 | 142.4 | 154.2 | 215.5 | 173.6 | 122.6 | 239.0 | 203.3 | 208.8 | 198.1 | 209.0 | 138.8 | 177.3 | 160.5 |
| Sr | 83.90 | 91.10 | 103.70 | 98.80 | 98.70 | 76.10 | 86.00 | 60.40 | 7.90 | 14.90 | 18.80 | 17.70 | 50.00 | 78.10 | 31.30 | 66.10 | 8.80 |

**Table 2.** *Cont.*

| Rock | Syenogranite | | | | | | | | | | | | | | | Alkali Feldspar Granite | |
|---|---|---|---|---|---|---|---|---|---|---|---|---|---|---|---|---|---|
| S. No. | 32 | 33 | 34 | 38 | 45 | 46 | 47 | 49 | 55 | 57 | 58 | 59 | 61 | 62 | 84 | 15 | 39 |
| Major oxides (wt%) | | | | | | | | | | | | | | | | | |
| $SiO_2$ | 76.03 | 75.46 | 76.46 | 76.21 | 71.66 | 73.41 | 72.25 | 73.59 | 72.88 | 74.43 | 72.36 | 74.95 | 71.68 | 75.12 | 74.02 | 78.68 | 77.18 |
| $TiO_2$ | 0.09 | 0.08 | 0.07 | 0.07 | 0.16 | 0.14 | 0.17 | 0.15 | 0.19 | 0.15 | 0.16 | 0.14 | 0.19 | 0.10 | 0.13 | 0.04 | 0.06 |
| $Al_2O_3$ | 12.59 | 12.95 | 12.58 | 12.87 | 14.19 | 13.43 | 13.63 | 13.58 | 13.89 | 13.27 | 15.4 | 13.51 | 13.95 | 13.14 | 13.49 | 12.01 | 12.5 |
| $Fe_2O_3$ * | 1.61 | 1.65 | 1.45 | 1.36 | 2.84 | 2.51 | 3.10 | 2.33 | 2.63 | 2.18 | 1.32 | 1.92 | 3.01 | 1.50 | 2.07 | 0.70 | 1.09 |
| MnO | 0.05 | 0.04 | 0.03 | 0.02 | 0.07 | 0.07 | 0.08 | 0.06 | 0.04 | 0.04 | 0.02 | 0.02 | 0.08 | 0.03 | 0.04 | 0.02 | 0.01 |
| MgO | 0.05 | 0.03 | 0.02 | 0.04 | 0.05 | 0.02 | 0.07 | 0.08 | 0.17 | 0.13 | 0.02 | 0.03 | 0.05 | 0.05 | 0.10 | 0.05 | 0.01 |
| CaO | 0.56 | 0.48 | 0.39 | 0.35 | 0.67 | 0.71 | 0.75 | 0.67 | 0.82 | 0.69 | 0.20 | 0.30 | 0.68 | 0.70 | 0.70 | 0.23 | 0.18 |
| $Na_2O$ | 4.67 | 4.81 | 4.83 | 4.77 | 5.18 | 4.88 | 5.32 | 5.03 | 4.77 | 4.59 | 4.69 | 4.24 | 5.49 | 4.65 | 4.96 | 4.66 | 4.85 |
| $K_2O$ | 4.34 | 4.49 | 4.16 | 4.31 | 5.17 | 4.8 | 4.61 | 4.48 | 4.57 | 4.50 | 5.80 | 4.87 | 4.87 | 4.68 | 4.47 | 3.62 | 4.12 |
| $P_2O_5$ | 0.01 | 0.02 | 0.01 | 0.01 | 0.01 | 0.01 | 0.02 | 0.02 | 0.04 | 0.03 | 0.03 | 0.02 | 0.01 | 0.02 | 0.02 | 0.01 | 0.02 |
| Total | 99.99 | 99.99 | 99.99 | 100 | 100 | 99.98 | 100 | 99.99 | 100 | 100 | 100 | 100 | 100 | 99.99 | 100 | 100 | 100 |
| CIPW norm | | | | | | | | | | | | | | | | | |
| Qz | 31.46 | 29.64 | 31.92 | 31.32 | 20.92 | 25.76 | 22.91 | 26.11 | 25.92 | 29.12 | 22.51 | 31.02 | 20.74 | 28.54 | 26.90 | 37.18 | 39.00 |
| Or | 25.65 | 26.53 | 24.58 | 25.47 | 30.55 | 28.37 | 27.24 | 26.47 | 27.01 | 26.59 | 34.27 | 28.78 | 28.78 | 28.72 | 26.42 | 39.43 | 32.96 |
| Ab | 39.52 | 40.70 | 40.87 | 40.36 | 43.83 | 41.29 | 44.44 | 42.56 | 40.36 | 38.84 | 39.69 | 35.88 | 44.64 | 39.35 | 41.97 | 21.39 | 24.35 |
| An | 0.57 | 0.48 | 0.36 | 0.98 | 0.20 | 0.56 | 0 | 1.24 | 2.99 | 2.31 | 0.80 | 1.36 | 0 | 0.63 | 1.34 | 1.08 | 1.04 |
| Cor | 0 | 0 | 0 | 0 | 0 | 0 | 0 | 0 | 0 | 0 | 1.11 | 0.77 | 0 | 0 | 0 | 0.03 | 0.17 |
| Di/en | 0.27 | 0.16 | 0.11 | 0.21 | 0.27 | 0.11 | 0.38 | 0.43 | 0.24 | 0.43 | 0 | 0 | 0.27 | 0.27 | 0.54 | 0 | 0 |
| Hy | 0 | 0 | 0 | 0 | 0 | 0 | 0 | 0 | 0.31 | 0.13 | 0.05 | 0.07 | 0 | 0 | 0 | 0.12 | 0.05 |

**Table 2.** *Cont.*

| | | | | | | | | | | | | | | | | | |
|---|---|---|---|---|---|---|---|---|---|---|---|---|---|---|---|---|---|
| Di/wo | 0.73 | 0.66 | 0.55 | 0.13 | 1.02 | 1.06 | 1.18 | 0.46 | 0 | 0 | 0 | 0 | 1.09 | 0.89 | 0.42 | 0 | 0 |
| Ac | 0 | 0 | 0 | 0 | 0 | 0.00 | 0.51 | 0 | 0 | 0 | 0 | 0 | 1.60 | 0 | 0 | 0 | 0.20 |
| Ilm | 0.11 | 0.09 | 0.06 | 0.04 | 0.15 | 0.15 | 0.17 | 0.13 | 0.09 | 0.09 | 0.04 | 0.04 | 0.17 | 0.06 | 0.09 | 0.04 | 0 |
| Hm | 1.61 | 1.65 | 1.45 | 1.36 | 2.84 | 2.51 | 2.93 | 2.33 | 2.63 | 2.18 | 1.32 | 1.92 | 2.46 | 1.50 | 2.07 | 0.70 | 0.02 |
| Ap | 0 | 0 | 0 | 0 | 0.02 | 0.02 | 0.05 | 0.05 | 0.09 | 0.07 | 0.07 | 0.05 | 0.02 | 0.05 | 0.05 | 0.02 | 1.09 |
| Tn | 0.08 | 0.09 | 0.09 | 0.12 | 0.20 | 0.15 | 0.20 | 0.20 | 0.36 | 0.26 | 0 | 0 | 0.25 | 0.16 | 0.21 | 0 | 0 |
| Ru | 0 | 0 | 0 | 0 | 0 | 0 | 0 | 0 | 0 | 0 | 0 | 0 | 0 | 0 | 0 | 0 | 0.12 |
| Elements (ppm) | | | | | | | | | | | | | | | | | |
| Ce | 79.00 | 72.60 | 64.10 | 57.10 | 214.0 | 221.9 | 184.4 | 164.2 | 107.2 | 102.2 | 67.80 | 32.60 | 184.3 | 76.80 | 88.00 | 28.30 | 52.10 |
| Th | 26.73 | 30.83 | 31.44 | 33.13 | 22.21 | 24.64 | 23.22 | 32.41 | 34.07 | 40.89 | 27.33 | 17.38 | 23.62 | 41.36 | 33.38 | 24.01 | 6.99 |
| U | 7.20 | 7.00 | 7.00 | 10.70 | | 6.40 | 15.30 | 7.50 | 7.70 | 8.30 | 10.20 | 3.60 | 10.00 | 15.20 | | 4.30 | 2.40 |
| Cr | 98.70 | 166.0 | 111.5 | 80.80 | 52.40 | 194.9 | 129.6 | 72.50 | 118.5 | 159.8 | 72.10 | 150.7 | 83.80 | 66.50 | 140.4 | 108.0 | 237.0 |
| Y | 65.80 | 60.50 | 57.70 | 44.60 | 44.10 | 46.10 | 47.40 | 51.70 | 40.30 | 35.70 | 43.70 | 37.80 | 49.10 | 51.40 | 51.20 | 33.30 | 55.60 |
| Nb | 145.2 | 183.6 | 169.4 | 162.0 | 102.1 | 103.5 | 117.9 | 124.9 | 78.47 | 67.43 | 164.5 | 75.13 | 125.6 | 117.5 | 125.3 | 137.7 | 76.02 |
| Zr | 211.5 | 225.5 | 219.8 | 213.9 | 562.5 | 589.1 | 646.7 | 361.6 | 280.0 | 237.6 | 463.6 | 369.4 | 552.8 | 214.4 | 291.7 | 184.5 | 89.40 |
| Ni | 2.50 | 1.00 | 1.20 | 0.20 | 1.50 | 0.30 | 7.80 | 0.30 | 1.00 | 0.10 | 1.50 | 0.70 | 2.60 | 1.30 | 1.20 | 1.10 | 3.80 |
| Ba | 91.70 | 88.30 | 76.40 | 73.90 | 187.8 | 141.2 | 224.0 | 234.6 | 257.4 | 232.2 | 335.0 | 316.0 | 218.3 | 141.5 | 178.5 | 81.40 | 43.90 |
| Pb | 8.80 | 13.70 | 15.20 | 4.80 | 5.30 | 3.90 | 5.00 | 5.40 | 6.10 | 5.40 | 5.30 | 3.80 | 9.10 | 6.50 | 9.90 | 2.20 | 3.00 |
| Rb | 175.6 | 191.4 | 205.3 | 204.0 | 140.5 | 136.2 | 183.2 | 195.80 | 204.2 | 212.8 | 208.3 | 151.8 | 201.0 | 265.2 | 205.0 | 172.0 | 231.7 |
| Sr | 12.40 | 9.30 | 10.00 | 6.50 | 38.20 | 24.00 | 48.00 | 72.00 | 73.40 | 63.90 | 137.8 | 74.00 | 41.10 | 30.70 | 44.10 | 7.60 | 3.10 |

**Table 2.** *Cont.*

| Rock | Alkali Feldspar Granite | | | Quartz Syenite | | | | | | | | | | | | | |
|---|---|---|---|---|---|---|---|---|---|---|---|---|---|---|---|---|---|
| S. No. | 41 | 42 | 43 | 22 | 23 | 26 | 44 | 48 | 50 | 51 | 56 | 63 | 64 | 67 | 75 | 76 | 68c |
| Major oxides (wt%) | | | | | | | | | | | | | | | | | |
| $SiO_2$ | 77.26 | 78.33 | 78.58 | 67.47 | 67.6 | 68.14 | 65.56 | 69.68 | 65.85 | 66.69 | 71.12 | 70.93 | 67.08 | 70.58 | 65.08 | 67.77 | 68.77 |
| $TiO_2$ | 0.06 | 0.06 | 0.05 | 0.26 | 0.27 | 0.29 | 0.26 | 0.21 | 0.38 | 0.36 | 0.22 | 0.22 | 0.27 | 0.20 | 0.32 | 0.30 | 0.26 |
| $Al_2O_3$ | 12.04 | 11.54 | 11.47 | 15.72 | 15.65 | 15.17 | 17.14 | 15.01 | 15.83 | 15.20 | 14.5 | 14.55 | 15.72 | 14.37 | 16.64 | 14.93 | 15.93 |
| $Fe_2O_3$ * | 1.73 | 1.68 | 1.36 | 4.05 | 4.04 | 4.24 | 3.90 | 3.19 | 5.09 | 5.19 | 3.05 | 3.10 | 4.29 | 3.25 | 4.71 | 4.68 | 3.83 |
| MnO | 0.06 | 0.03 | 0.02 | 0.12 | 0.09 | 0.10 | 0.10 | 0.08 | 0.15 | 0.14 | 0.07 | 0.06 | 0.10 | 0.08 | 0.11 | 0.10 | 0.03 |
| MgO | 0.02 | 0.02 | 0.02 | 0.08 | 0.09 | 0.15 | 0.13 | 0.06 | 0.09 | 0.08 | 0.20 | 0.16 | 0.08 | 0.07 | 0.17 | 0.03 | 0.11 |
| CaO | 0.23 | 0.25 | 0.21 | 1.21 | 1.22 | 1.21 | 1.54 | 0.86 | 1.19 | 1.14 | 0.94 | 0.82 | 1.18 | 0.80 | 1.68 | 1.03 | 0.27 |
| $Na_2O$ | 4.31 | 4.34 | 4.61 | 6.18 | 6.11 | 5.96 | 6.49 | 5.67 | 5.82 | 5.43 | 5.20 | 5.56 | 6.42 | 5.78 | 6.68 | 5.90 | 4.83 |
| $K_2O$ | 4.29 | 3.77 | 3.66 | 4.88 | 4.90 | 4.68 | 4.86 | 5.23 | 5.57 | 5.75 | 4.66 | 4.55 | 4.83 | 4.86 | 4.57 | 5.20 | 5.96 |
| $P_2O_5$ | 0.01 | 0.01 | 0.01 | 0.02 | 0.03 | 0.05 | 0.03 | 0.01 | 0.03 | 0.02 | 0.05 | 0.04 | 0.03 | 0.02 | 0.04 | 0.01 | 0.03 |
| Total | 100 | 100 | 99.98 | 99.99 | 100 | 99.99 | 100 | 100 | 100 | 100 | 100 | 99.99 | 100 | 100 | 100 | 99.95 | 100 |
| CIPW norm | | | | | | | | | | | | | | | | | |
| Qz | 37.02 | 38.05 | 39.04 | 11.30 | 11.74 | 13.94 | 6.70 | 15.67 | 9.18 | 11.76 | 11.27 | 19.83 | 9.99 | 17.9 | 6.32 | 12.89 | 17.21 |
| Or | 35.32 | 35.17 | 35.59 | 28.84 | 28.96 | 27.66 | 28.72 | 30.91 | 32.92 | 33.98 | 27.54 | 26.89 | 28.54 | 28.72 | 27.01 | 30.73 | 35.22 |
| Ab | 23.35 | 22.28 | 21.63 | 52.29 | 51.70 | 50.43 | 54.92 | 47.98 | 49.25 | 45.95 | 44.00 | 47.05 | 53.97 | 46.86 | 56.52 | 47.84 | 40.87 |
| An | 1.47 | 1.72 | 1.62 | 0.74 | 0.80 | 0.82 | 3.28 | 0.06 | 0.62 | 0.12 | 2.46 | 1.31 | 0 | 0 | 1.92 | 0 | 1.14 |
| Cor | 0.84 | 0.87 | 0 | 0 | 0 | 0 | 0 | 0 | 0 | 0 | 0 | 0 | 0 | 0 | 0 | 0 | 1.11 |
| Di/en | 0 | 0 | 0 | 0.43 | 0.48 | 0.81 | 0.70 | 0.32 | 0.48 | 0.43 | 11.07 | 0.86 | 0.43 | 0.38 | 0.91 | 0.16 | 0 |
| Hy | 0.11 | 0.11 | 0.11 | 0 | 0 | 0 | 0 | 0 | 0 | 0 | 0 | 0 | 0 | 0 | 0 | 0 | 0.27 |

**Table 2.** *Cont.*

| | | | | | | | | | | | | | | | | | |
|---|---|---|---|---|---|---|---|---|---|---|---|---|---|---|---|---|---|
| Di/wo | 0 | 0 | 0 | 1.73 | 1.60 | 1.34 | 1.15 | 1.38 | 1.56 | 1.73 | 0 | 0.36 | 1.90 | 1.24 | 1.79 | 1.75 | 0 |
| Ac | 0.07 | 0.06 | 0.34 | 0 | 0 | 0 | 0 | 0 | 0 | 0 | 0 | 0.00 | 0.31 | 1.81 | 0.00 | 1.83 | 0 |
| Ilm | 0 | 0 | 0.34 | 0.26 | 0.19 | 0.21 | 0.21 | 0.17 | 0.32 | 0.30 | 0.15 | 0.13 | 0.21 | 0.17 | 0.24 | 0.21 | 0.06 |
| Hm | 0.11 | 0.06 | 0.04 | 4.05 | 4.04 | 4.24 | 3.90 | 3.19 | 5.09 | 5.19 | 3.05 | 3.10 | 4.18 | 2.63 | 4.71 | 4.05 | 3.83 |
| Ap | 1.73 | 1.68 | 1.24 | 0.05 | 0.07 | 0.12 | 0.07 | 0.02 | 0.07 | 0.05 | 0.12 | 0.09 | 0.07 | 0.05 | 0.09 | 0.02 | 0.07 |
| Tn | 0 | 0 | 0 | 0.31 | 0.41 | 0.44 | 0.36 | 0.29 | 0.52 | 0.50 | 0.35 | 0.37 | 0.39 | 0.27 | 0.48 | 0.46 | 0 |
| Ru | 0 | 0.06 | 0.07 | 0 | 0 | 0 | 0 | 0 | 0 | 0 | 0 | 0 | 0 | 0 | 0 | 0 | 0.29 |
| Elements (ppm) | | | | | | | | | | | | | | | | | |
| Ce | 97.70 | 62.10 | 38.20 | 211.2 | 194.5 | 213.9 | 107.7 | 301.9 | 326.9 | 527.7 | 120.6 | 141.4 | 204.4 | 189.4 | 239.7 | 469.7 | 78.70 |
| Th | 16.70 | 14.50 | 4.38 | 17.50 | 13.26 | 17.09 | 10.48 | 34.90 | 22.47 | 36.55 | 30.67 | 28.08 | 22.24 | 24.81 | 17.10 | 30.84 | 23.89 |
| U | 3.50 | 3.50 | 2.20 | 5.40 | 3.40 | 6.00 | 3.20 | 7.30 | 7.70 | 11.0 | 6.90 | 7.20 | 9.60 | 6.80 | 7.00 | 8.70 | 5.70 |
| Cr | 140.6 | 198.2 | 121.9 | 65.60 | 54.90 | 88.70 | 53.00 | 80.90 | 30.60 | 50.90 | 121.9 | 124.5 | 74.50 | 85.70 | 39.10 | 83.80 | 85.00 |
| Y | 42.60 | 37.90 | 50.60 | 37.50 | 45.10 | 44.70 | 31.80 | 58.90 | 42.70 | 59.40 | 37.80 | 46.60 | 43.30 | 46.90 | 40.70 | 56.50 | 56.00 |
| Nb | 102.4 | 80.63 | 60.07 | 99.08 | 100.8 | 113.17 | 79.39 | 148.9 | 98.97 | 179.5 | 75.22 | 125.3 | 126.17 | 114.9 | 95.90 | 168.5 | 245.3 |
| Zr | 154.7 | 113.7 | 68.40 | 558.9 | 584.4 | 603.1 | 552.1 | 764.1 | 311.4 | 393.4 | 348.1 | 467.7 | 666.3 | 638.1 | 749.8 | 241.9 | 875.6 |
| Ni | 0.30 | 1.70 | 1.40 | 1.30 | 1.60 | 0.20 | 2.60 | 1.10 | 2.80 | 0.10 | 0.50 | 0.60 | 0.80 | 1.70 | 2.10 | 1.60 | 1.60 |
| Ba | 57.80 | 39.30 | 50.70 | 577.9 | 537.2 | 454.5 | 829.2 | 153.5 | 453.0 | 131.4 | 328.3 | 292.8 | 567.6 | 271.3 | 697.6 | 129.0 | 277.7 |
| Pb | 10.90 | 5.30 | 3.90 | 7.50 | 6.20 | 6.80 | 5.80 | 6.10 | 13.00 | 5.80 | 4.00 | 4.80 | 7.70 | 6.00 | 5.60 | 6.60 | 11.10 |
| Rb | 244.2 | 202.9 | 206.0 | 129.8 | 109.5 | 130.0 | 115.6 | 167.70 | 135.3 | 160.3 | 208.9 | 197.2 | 130.0 | 179.5 | 101.7 | 162.4 | 192.5 |
| Sr | 5.50 | 7.50 | 5.50 | 116.0 | 104.2 | 122.4 | 231.8 | 27.60 | 80.10 | 29.40 | 94.70 | 90.60 | 109.4 | 55.20 | 220.8 | 17.50 | 96.40 |

\* CIPW norm (Cross, Iddings, Pirrson and Washington) is a useful scheme because abundances of normative minerals are required for a proper rock classification.

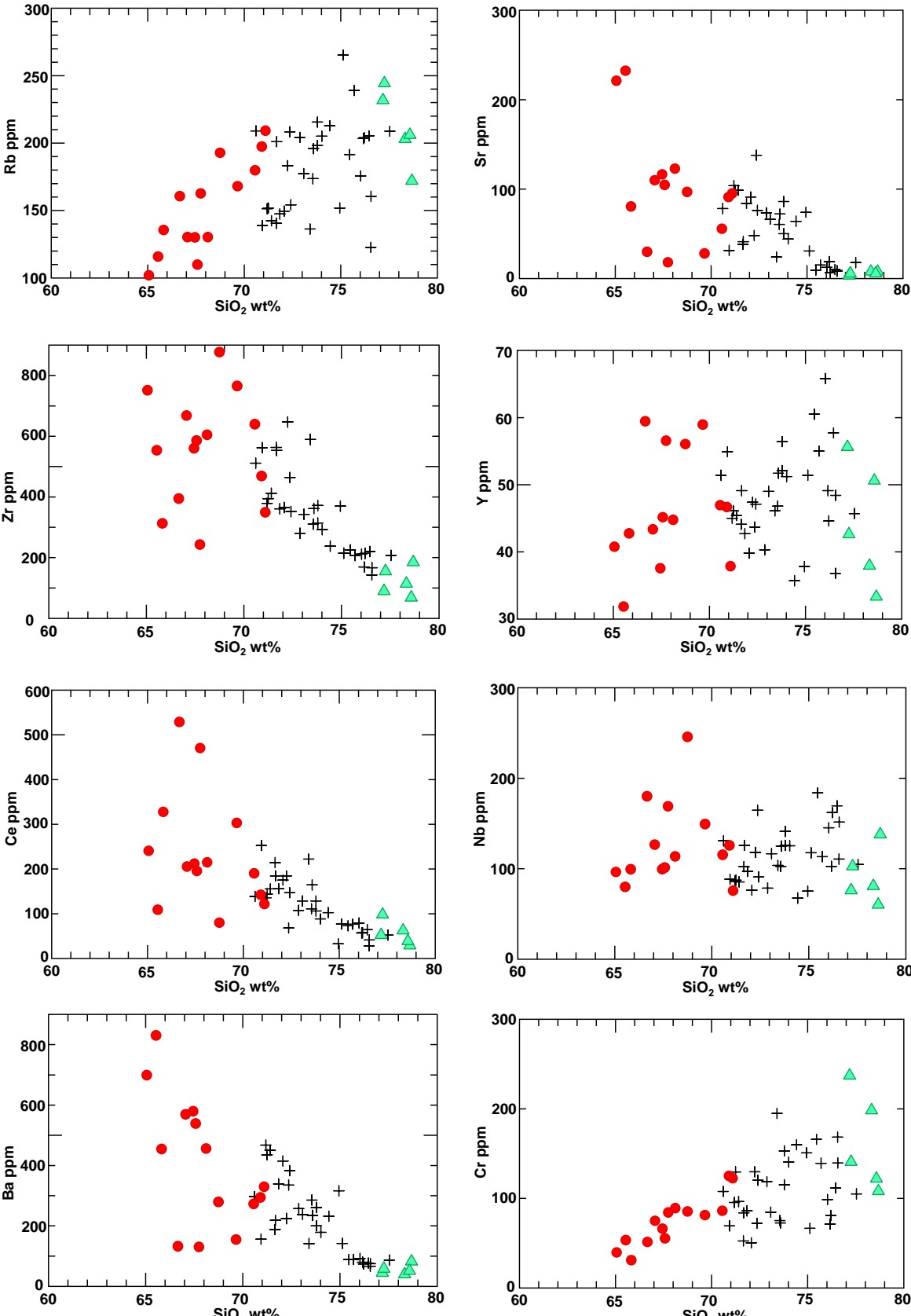

**Figure 3.** Harker variation diagrams of major oxides (wt%) and some trace elements (ppm) versus SiO₂ wt%. Symbols are the same in the following diagrams (+ syenogranite, ▲ alkali feldspar granite and ● quartz syenite).

Many chemical classifications of igneous rocks have been presented by many authors using different geochemical parameters. The $SiO_2$ versus ($Na_2O + K_2O$) (TAS) classification diagram [40] (Figure 4a) reveals that six of the samples plot in alkali feldspar granite, while the syenogranite and alkali feldspar granite samples plot in the alkali feldspar granite field.

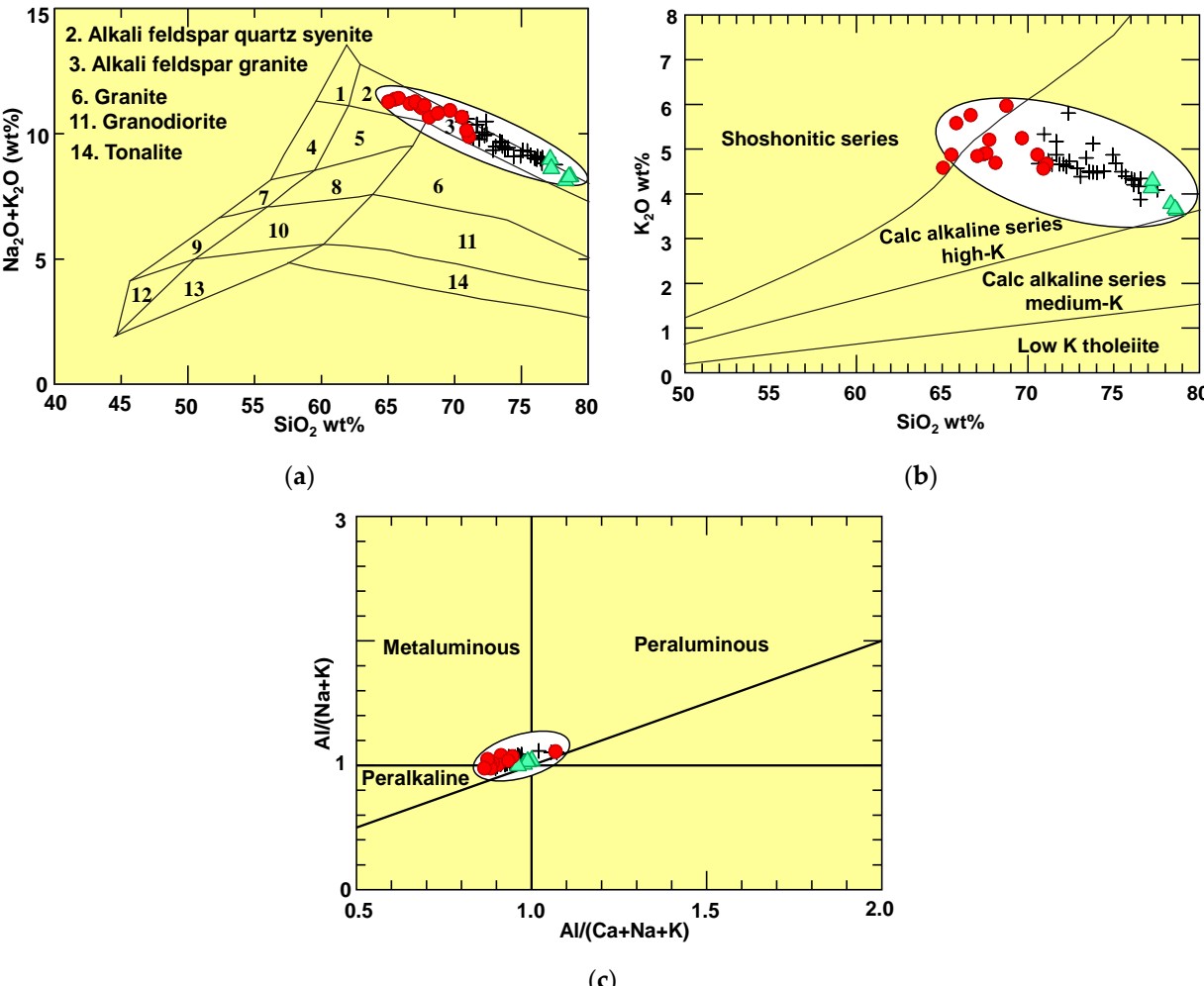

**Figure 4.** Geochemical nomenclature and magma type diagrams for the studied granitoids. (**a**) The total alkalis versus silica (TAS) diagram after Middlemost [40]; (**b**) Rickwood [41] diagram of $SiO_2$ versus $K_2O$; (**c**) Molar $Al_2O_3/(CaO + Na_2O + K_2O)$ versus $Al_2O_3/(Na_2O + K_2O)$ diagram after Maniar and Piccoli [42]. Symbols are the same in the following diagrams (**+** syenogranite, ▲ alkali feldspar granite and ● quartz syenite).

For sub-alkaline rocks, Rickwood [41] used the $K_2O$–$SiO_2$ discrimination diagram (Figure 4b) to differentiate between shoshonite series, high K calc–alkaline series, medium K calc–alkaline series and low K tholeiites. The studied quartz syenite samples straddle the boundary line between the high shoshonitic and high K calc–alkaline series fields, whereas the syenogranite and alkali feldspar granite plot in the high K calc–alkaline series field.

Maniar and Piccoli [42] used $(Al_2O_3)/(Na_2O + K_2O + CaO)$ versus $(Al_2O_3)/(Na_2O + K_2O)$ to distinguish between peraluminous, metaluminous and peralkaline rocks (Figure 4c). The figure reveals that the studied rocks fall mainly in the lowermost right corner of the metaluminous field, except some samples that are plotted in the peralkaline field, while three syenogranite and one quartz syenite samples are plotted in the peraluminous field.

The tectonic setting of the studied rocks can be inferred using binary discrimination diagrams. Pearce et al. [43] used Rb–(Nb + Y) and Nb–Y discrimination diagrams to distinguish between the four tectonic regimes, including volcanic arc granites (VAG),

syn-collision granites (syn-COLG), ocean ridge granites (ORG) and within plate granites (WPG) for the granite rocks. The studied rock samples plot in the within-plate granite field (Figure 5a,b). This reveals that these rocks were emplaced in a within-plate regime under an extensional (Anorogenic) environment. Maniar and Piccoli [42] published some diagrams to discriminate the tectonic setting of igneous rocks. The $Al_2O_3$–$SiO_2$ diagram (Figure 5c) discriminates post-orogenic granites (POG) from the other granitic tectonic settings. Due to low $SiO_2$ (wt%) content, most quartz syenites are not plotted on the diagram, which also shows that almost all other analyzed samples plot in the post orogenic granites field. Whalen et al. [44] used the binary diagrams $Zr + Nb + Ce + Y$ (ppm) versus $Fe_2O_3/MgO$ to discriminate A-type granite from highly and normal fractionated S/I-type granites (Figure 5d). From these figures, it is noted that all the samples fall in A-type granites. Sylvester [45] mentioned that highly fractionated post-collision calc-alkaline granites are similar to A-type granites.

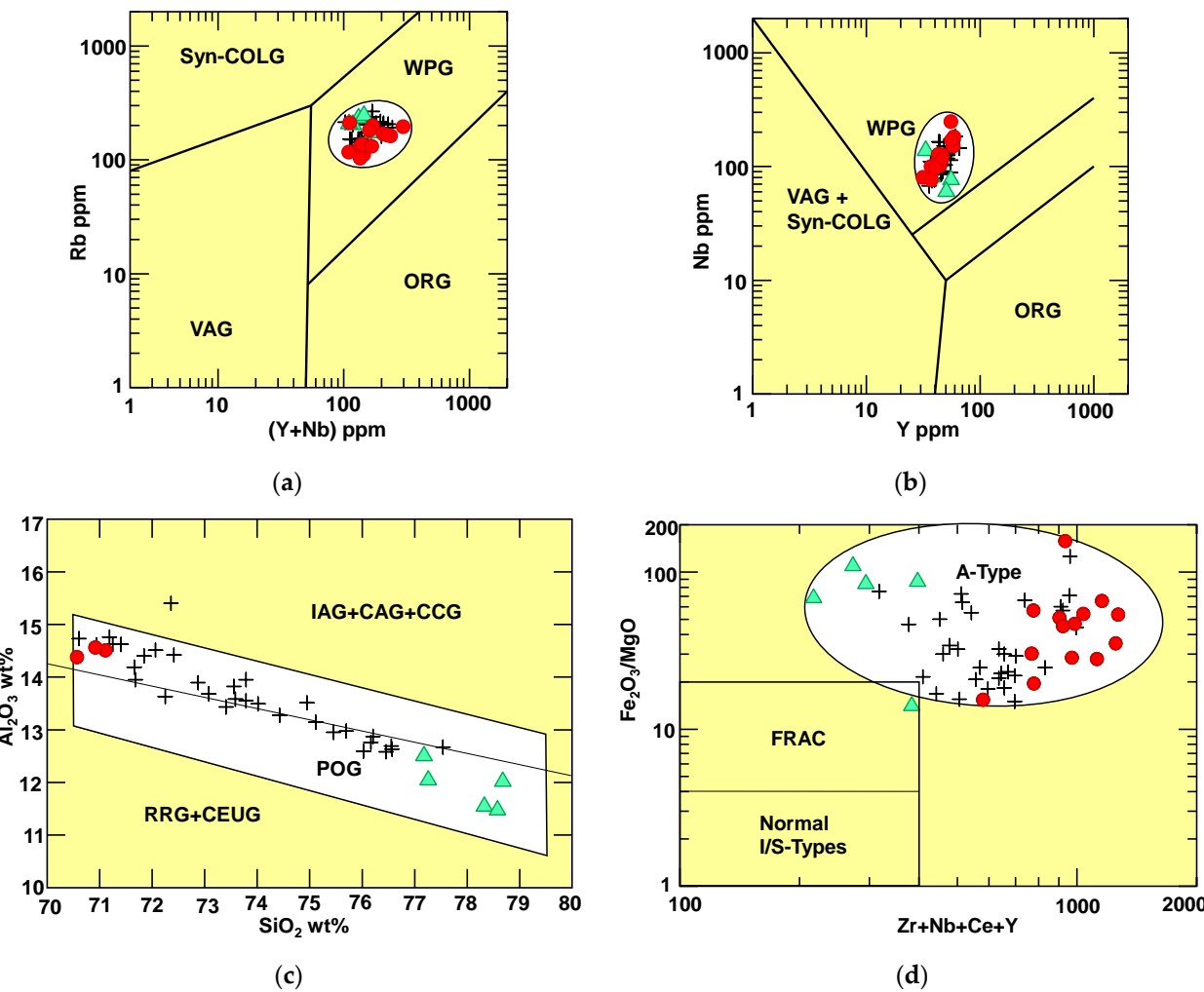

**Figure 5.** Tectonic setting diagrams for the studied granites and quartz syenite. (**a**,**b**) Rb versus (Y + Nb) diagrams after Pearce et al. [42]; (**c**) $Al_2O_3$ versus $SiO_2$ diagram after Maniar and Piccoli [42]; (**d**) $Zr + Nb + Ce + Y$ versus $Fe_2O_3/MgO$ diagram after Whalen et al. [44]. Symbols are the same in the following diagrams (+ syenogranite, ▲ alkali feldspar granite and ● quartz syenite).

### 4.3. Petrogenesis of Granitoids

Metasomatized granites are usually characterized by extreme enrichment of Rb, similar to Nigerian granites [45], but magmatic differentiated alkaline syenites and granites usually have moderate Rb levels, as in those of the Ras ed Dome complex [46].

Rb, Sr, and Ba are the most helpful elements to evaluate the origin of granite rocks either from partial melting or fractional crystallization. Their behavior in the granitic system is strongly controlled by K-feldspar, plagioclase and mica. K/Rb ratio is a petrogenetic indicator that usually decreases with increasing magmatic fractionation.

$K_2O$ content reflects the degree of Rb enrichment in the magmatic rocks. On the $K_2O$ (wt%) versus Rb (ppm) diagram, quartz syenite and very few syenogranite samples plot within the field of the Ras ed Dome ring complex, Sudan [47]. Moreover, the studied samples show a general horizontal trend that deviates from the major magmatic trends and plot close to the Shaw [48] pegmatitic–hydrothermal trend (Figure 6a), except for the illite and albite content of the deviated samples of syenogranite and alkali feldspar granite, interpreted by Vidal et al. [46] to reflect the role of post-magmatic auto-metasomatic alterations.

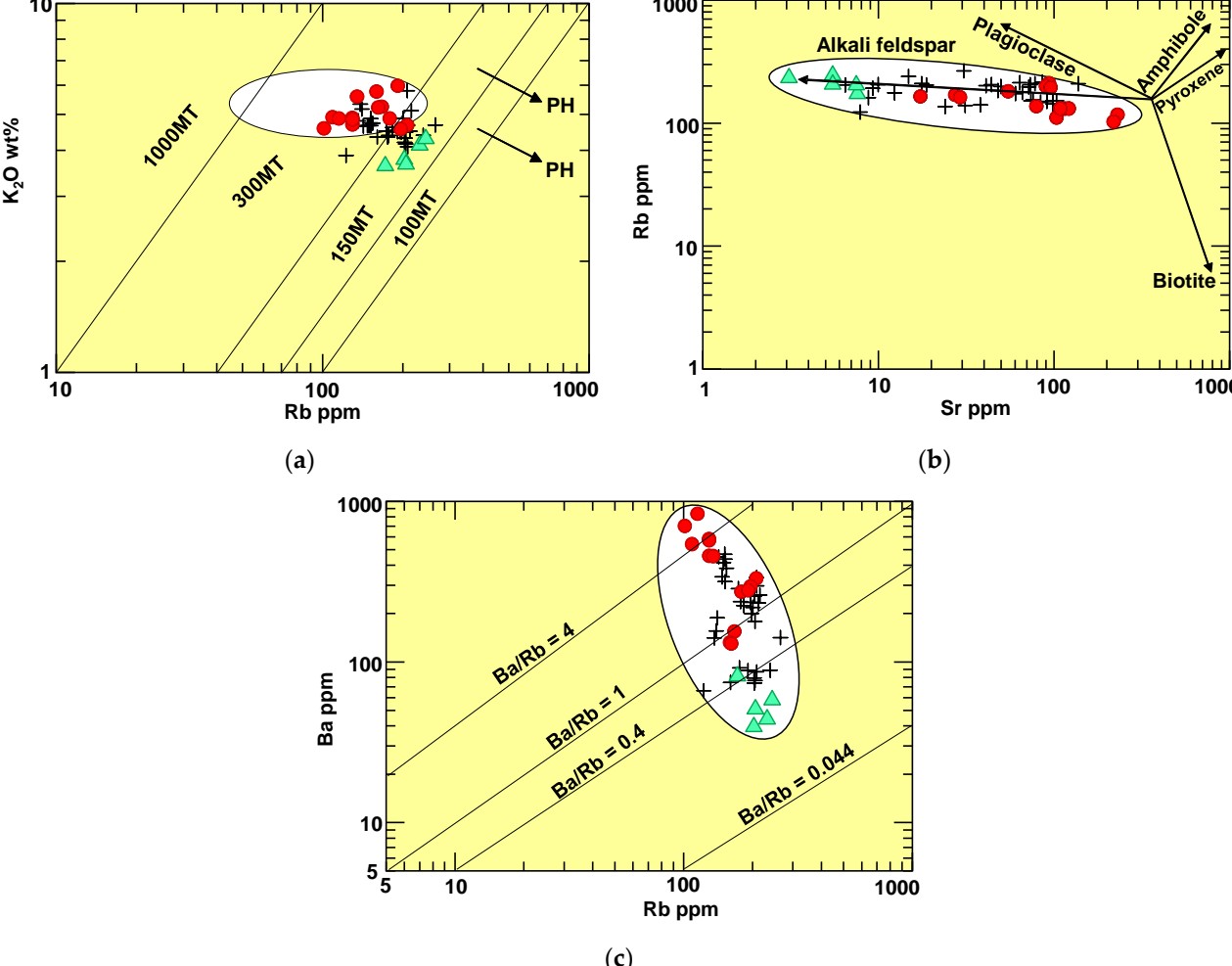

**Figure 6.** Petrogenesis diagrams of the studied granites and quartz syenite. (**a**) $K_2O$ versus Rb diagram, MT: magmatic trend, PH: pegmatitic hydrothermal trend after Shaw [48]. Shaded area is the field of the Ras ed Dome ring complex, Sudan after O'Halloran [46]; (**b**) Sr versus Rb diagram for the studied samples; (**c**) Ba versus Rb diagram after Mason [49]. Symbols are the same in the following diagrams (**+** syenogranite, ▲ alkali feldspar granite and ● quartz syenite).

Another helpful relationship is Rb versus Sr (Figure 6b), which further asserts the role of K-feldspar fractionation in the development of alkali feldspar granite from syenogranite. This result agrees with their hypersolvus character and the high proportion of microperthites.

The average Ba/Rb ratio for the crust is about 4.4 [49]. Figure 6c shows that the studied alkali feldspar granite and a few syenogranite samples plot between Ba/Rb values of 0.4 to 0.044, which indicates Rb enrichment. Most quartz syenite samples plot along the

Ba/Rb ratio line of about 4.4, which indicates enrichment of Ba over Rb. Most syenogranite samples plot between 0.4 and 4.4, which indicates Ba enrichment and/or depletion of Rb.

### 4.4. Spider Diagrams

The concentrations of some trace elements are normalized to a primitive mantle value of Sun and McDonough [50] in Figure 7a–c. Quartz syenite, syenogranite and alkali feldspar granite show enrichment of large-ion lithophile elements (LILE; especially Rb) and high field-strength elements (HFSE; Zr, Nb) and depletion (troughs) of Ba, Sr and Ti. The depletion of Ti is ascribed to fractionation of titanomagnetite.

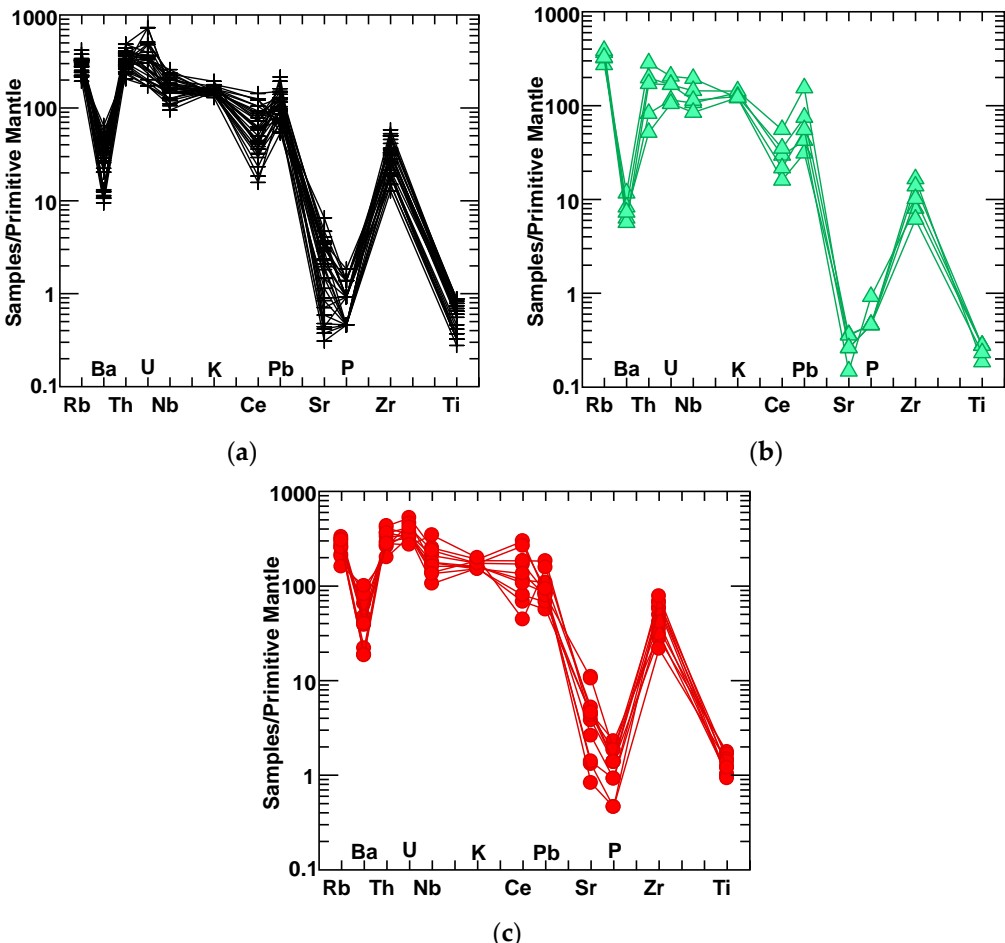

**Figure 7.** Normalized multi-element patterns of the studied granitoids: (**a**) syenogranite; (**b**) alkali feldspar granite; (**c**) quartz syenite. Symbols are the same in the following diagrams (**+** syenogranite, ▲ alkali feldspar granite and ● quartz syenite).

### 4.5. Geochemical Features of U and Th in Granitoids

The variations of U, Th, U/Th ratio and Zr are shown in Figure 8. U and Th show notable variations, with a slight correlation with $SiO_2$, especially in quartz syenite and syenogranite. The alkali feldspar granite samples show marked depletion in both U and Th content relative to others. Better positive correlation between U and Th may indicate that U is mainly located in Th-rich accessory minerals such as uranothorite, zircon, monazite and allanite [33]. The Th/U ratio shows marked fluctuation within each rock type with increasing U content, but there is a relatively positive correlation with increasing Th concentration in granitoids. U and Th versus Zr relation shows a relatively negative correlation in quartz syenite and syenogranite but a positive correlation in alkali feldspar granite, which reflects that zircon is one of the most important U–Th bearing minerals in alkali feldspar granite.

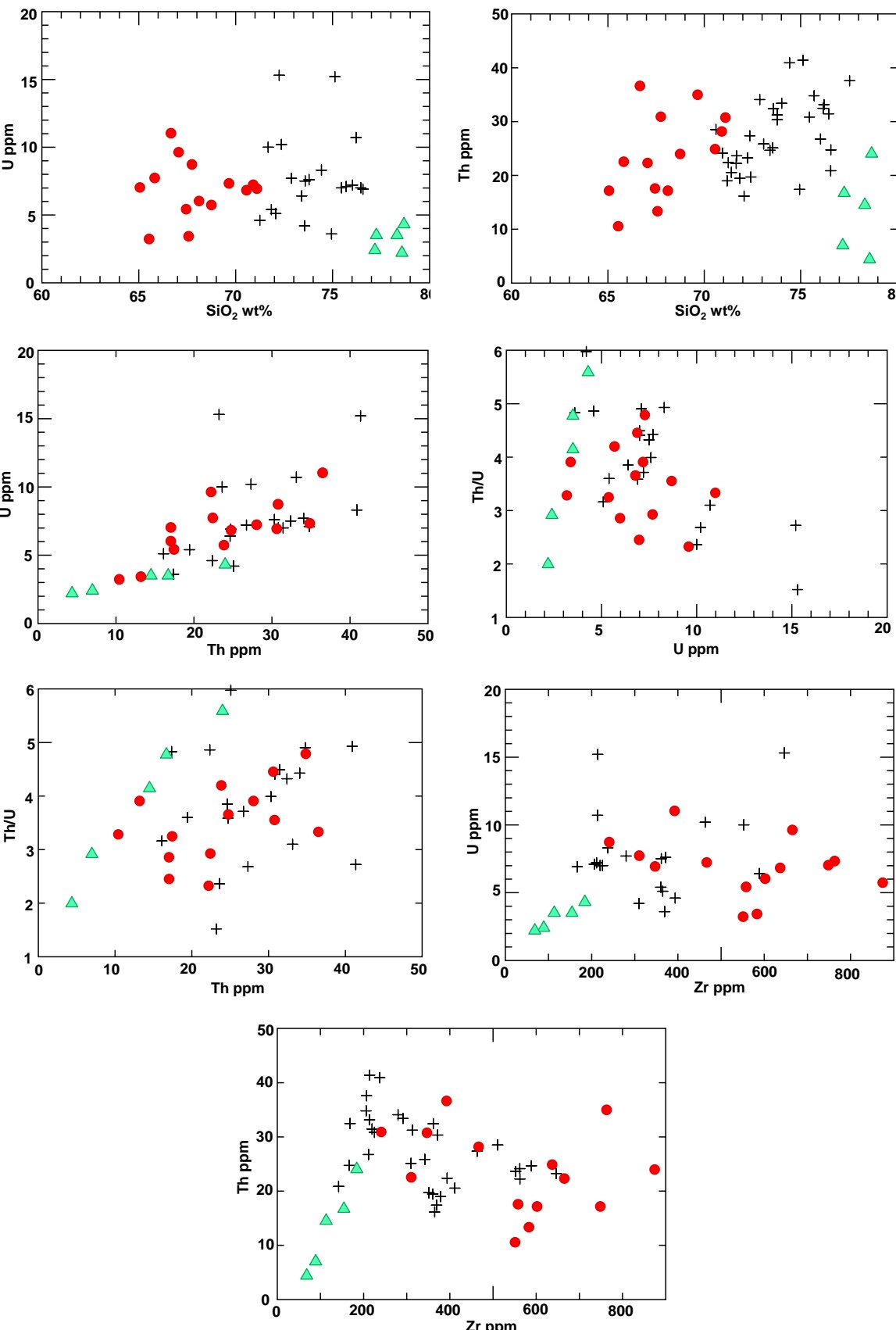

**Figure 8.** Binary variation diagrams for Nikeiba granitoids. Symbols are the same in the following diagrams (+ syenogranite, ▲ alkali feldspar granite and ● quartz syenite).

### 4.6. Radiological Assessment

4.6.1. Activity Concentration

Table 3 exhibits the $^{238}$U, $^{232}$Th and $^{40}$K activity concentrations of the granitoid samples (syenogranite, alkali feldspar granite and quartz syenite). The $^{238}$U, $^{232}$Th and $^{40}$K activity concentration average values exceed the worldwide averages of 33, 45 and 412, respectively [29]. As explained in the studied granitoid samples, the highest average $A_U$ of $131 \pm 49$ Bq·kg$^{-1}$ was observed in the syenogranite samples and is higher than the worldwide average value (33 Bq·kg$^{-1}$), while the lowest average value of $65 \pm 16$ Bq·kg$^{-1}$ was detected in the alkali feldspar granite samples. $A_U$ ranged from 47 to 184 Bq·kg$^{-1}$ for syenogranite samples, from 40 to 91 Bq·kg$^{-1}$ for alkali feldspar granite samples and from 52 to 111 Bq·kg$^{-1}$ for quartz syenite samples. The average values of $A_{Th}$ are $164 \pm 35$, $69 \pm 17$ and $105 \pm 13$ Bq·kg$^{-1}$ for syenogranite, alkali feldspar granite and quartz syenite samples, respectively, which are greater by a factor of 4, 1.5 and 2 than the 45 Bq·kg$^{-1}$ suggested worldwide average value. The maximum values of $A_{Th}$ are 229, 97 and 122 Bq·kg$^{-1}$ for syenogranite, alkali feldspar granite and quartz syenite, respectively, while the minimum values are 122, 41 and 84 Bq·kg$^{-1}$ for syenogranite, alkali feldspar granite and quartz syenite, respectively. The average values of $A_K$ ranged from 955 Bq·kg$^{-1}$ for alkali feldspar granite samples to 1402 Bq·kg$^{-1}$ for syenogranite samples. The lowest registered values of $A_K$ are 1033, 470 and 970 Bq·kg$^{-1}$, for syenogranite, alkali feldspar granite and quartz syenite samples, respectively. Moreover, the highest $A_K$ values are 1753, 1127, and 1690 Bq·kg$^{-1}$ for syenogranite, alkali feldspar granite and quartz syenite samples, respectively.

**Table 3.** Activity concentrations of $^{238}$U, $^{232}$Th and $^{40}$K (Bq·kg$^{-1}$) and the associated environmental hazard variables in the granitoids (syenogranite, alkali feldspar granite and quartz syenite).

| Sample | $^{238}$U Bq·kg$^{-1}$ | $^{232}$Th Bq·kg$^{-1}$ | $^{40}$K Bq·kg$^{-1}$ | $D_{air}$ (nGy/h) | $AED_{out}$ (mSv/y) | $AED_{in}$ (mSv/y) | AGDE (mSv/y) | $ELCR_{out}$ | $ELCR_{in}$ | $ELCR_t$ |
|---|---|---|---|---|---|---|---|---|---|---|
| | | | | | Syenogranite | | | | | |
| S1 | 75 | 122 | 1033 | 150.7 | 0.18 | 0.7 | 1.07 | 0.6 | 2.6 | 3.2 |
| S2 | 158 | 229 | 1690 | 280.4 | 0.34 | 1.4 | 1.97 | 1.2 | 4.8 | 6.0 |
| S3 | 100 | 155 | 1315 | 193.8 | 0.24 | 1.0 | 1.37 | 0.8 | 3.3 | 4.2 |
| S4 | 106 | 145 | 1628 | 203.3 | 0.25 | 1.0 | 1.45 | 0.9 | 3.5 | 4.4 |
| S5 | 184 | 174 | 1753 | 262.1 | 0.32 | 1.3 | 1.85 | 1.1 | 4.5 | 5.6 |
| S6 | 47 | 162 | 1221 | 169.8 | 0.21 | 0.8 | 1.21 | 0.7 | 2.9 | 3.6 |
| S7 | 116 | 133 | 1189 | 182.8 | 0.22 | 0.9 | 1.29 | 0.8 | 3.1 | 3.9 |
| S8 | 162 | 207 | 1440 | 258.6 | 0.32 | 1.3 | 1.82 | 1.1 | 4.4 | 5.5 |
| S9 | 184 | 125 | 1252 | 211.6 | 0.26 | 1.0 | 1.48 | 0.9 | 3.6 | 4.5 |
| S10 | 177 | 185 | 1502 | 254.8 | 0.31 | 1.2 | 1.79 | 1.1 | 4.4 | 5.5 |
| Ave | 131 | 164 | 1402 | 217 | 0.27 | 1.06 | 1.53 | 0.93 | 3.72 | 4.65 |
| SD | 49 | 35 | 239 | 44 | 0.05 | 0.22 | 0.31 | 0.19 | 0.76 | 0.95 |
| Min | 47 | 122 | 1033 | 151 | 0.18 | 0.74 | 1.07 | 0.65 | 2.59 | 3.23 |
| Max | 184 | 229 | 1753 | 280 | 0.34 | 1.38 | 1.97 | 1.20 | 4.81 | 6.02 |

<div align="center">Table 3. <em>Cont.</em></div>

| Sample | $^{238}$U Bq·kg$^{-1}$ | $^{232}$Th Bq·kg$^{-1}$ | $^{40}$K Bq·kg$^{-1}$ | $D_{air}$ (nGy/h) | $AED_{out}$ (mSv/y) | $AED_{in}$ (mSv/y) | AGDE (mSv/y) | $ELCR_{out}$ | $ELCR_{in}$ | $ELCR_t$ |
|---|---|---|---|---|---|---|---|---|---|---|
| | | | | Alkali feldspar granite | | | | | | |
| AF1 | 40 | 41 | 1002 | 84 | 0.10 | 0.41 | 0.61 | 0.36 | 1.45 | 1.81 |
| AF2 | 91 | 72 | 1002 | 127 | 0.16 | 0.62 | 0.90 | 0.54 | 2.18 | 2.72 |
| AF3 | 62 | 57 | 470 | 82 | 0.10 | 0.40 | 0.58 | 0.35 | 1.41 | 1.76 |
| AF4 | 65 | 84 | 1096 | 126 | 0.15 | 0.62 | 0.90 | 0.54 | 2.16 | 2.70 |
| AF5 | 79 | 67 | 1002 | 118 | 0.14 | 0.58 | 0.84 | 0.51 | 2.02 | 2.53 |
| AF6 | 52 | 97 | 876 | 118 | 0.15 | 0.58 | 0.84 | 0.51 | 2.03 | 2.54 |
| AF7 | 61 | 76 | 939 | 113 | 0.14 | 0.55 | 0.80 | 0.48 | 1.93 | 2.42 |
| AF8 | 86 | 85 | 1127 | 138 | 0.17 | 0.68 | 0.98 | 0.59 | 2.36 | 2.95 |
| AF9 | 51 | 51 | 1033 | 96 | 0.12 | 0.47 | 0.69 | 0.41 | 1.66 | 2.07 |
| AF10 | 68 | 60 | 1002 | 108 | 0.13 | 0.53 | 0.77 | 0.47 | 1.86 | 2.33 |
| Ave | 65 | 69 | 955 | 111 | 0.14 | 0.54 | 0.79 | 0.48 | 1.91 | 2.38 |
| SD | 16 | 17 | 184 | 18 | 0.02 | 0.09 | 0.13 | 0.08 | 0.32 | 0.39 |
| Min | 40 | 41 | 470 | 82 | 0.10 | 0.40 | 0.58 | 0.35 | 1.41 | 1.76 |
| Max | 91 | 97 | 1127 | 138 | 0.17 | 0.68 | 0.98 | 0.59 | 2.36 | 2.95 |
| | | | | Quartz syenite | | | | | | |
| Q1 | 59 | 84 | 1534 | 141 | 0.17 | 0.69 | 1.01 | 0.60 | 2.42 | 3.02 |
| Q2 | 64 | 122 | 1690 | 173 | 0.21 | 0.85 | 1.24 | 0.74 | 2.96 | 3.70 |
| Q3 | 111 | 117 | 1158 | 169 | 0.21 | 0.83 | 1.20 | 0.73 | 2.91 | 3.64 |
| Q4 | 98 | 97 | 1064 | 148 | 0.18 | 0.72 | 1.04 | 0.63 | 2.53 | 3.17 |
| Q1 | 52 | 102 | 1502 | 147 | 0.18 | 0.72 | 1.06 | 0.63 | 2.52 | 3.15 |
| Q2 | 59 | 93 | 1440 | 143 | 0.18 | 0.70 | 1.03 | 0.61 | 2.45 | 3.07 |
| Q3 | 74 | 111 | 1377 | 158 | 0.19 | 0.77 | 1.13 | 0.68 | 2.71 | 3.39 |
| Q4 | 86 | 121 | 1033 | 155 | 0.19 | 0.76 | 1.10 | 0.67 | 2.66 | 3.33 |
| Q1 | 78 | 97 | 1189 | 143 | 0.18 | 0.70 | 1.02 | 0.61 | 2.46 | 3.07 |
| Q2 | 74 | 104 | 970 | 137 | 0.17 | 0.67 | 0.97 | 0.59 | 2.35 | 2.94 |
| Ave | 76 | 105 | 1296 | 151 | 0.19 | 0.74 | 1.08 | 0.65 | 2.60 | 3.25 |
| SD | 19 | 13 | 245 | 12 | 0.01 | 0.06 | 0.09 | 0.05 | 0.21 | 0.26 |
| Min | 52 | 84 | 970 | 137 | 0.17 | 0.67 | 0.97 | 0.59 | 2.35 | 2.94 |
| Max | 111 | 122 | 1690 | 173 | 0.21 | 0.85 | 1.24 | 0.74 | 2.96 | 3.70 |

The skewness values characterize the asymmetric distribution following basic data analysis of radionuclide activity concentrations, with positive values indicating that the asymmetric distribution tail is expanded towards positive values and negative values indicating that the tail is expanded to negative values. As seen in Table 4, the skewness values of $A_U$ are + ve values in alkali feldspar granite and quartz syenite samples while −ve values in the syenogranite samples.

**Table 4.** Statistics for the corresponding data for $^{238}$U, $^{232}$Th and $^{40}$K activity concentrations in the granitoids.

| St. Par. | Syenogranite | | | Alkali Feldspar Granite | | | Quartz Syenite | | |
|---|---|---|---|---|---|---|---|---|---|
| | $^{238}$U | $^{232}$Th | $^{40}$K | $^{238}$U | $^{232}$Th | $^{40}$K | $^{238}$U | $^{232}$Th | $^{40}$K |
| Average | 131 | 164 | 1402 | 65 | 69 | 955 | 76 | 105 | 1296 |
| SD | 49 | 35 | 239 | 16 | 17 | 184 | 19 | 13 | 245 |
| Skewness | −0.4 | 0.6 | 0.1 | 0.2 | −0.0001 | −2.3 | 0.7 | −0.1 | 0.2 |
| Kurtosis | −1.1 | −0.5 | −1.2 | −0.7 | −0.8 | 6.4 | −0.1 | −1.0 | −1.4 |
| GM | 121 | 160 | 1384 | 160 | 67 | 104 | 1384 | 932 | 1275 |

Thus, $A_U$ exhibits positive asymmetry in alkali feldspar granite and quartz syenite samples, whereas the syenogranite sample distribution demonstrates negative asymmetry. Moreover, a positive distribution is found for $A_{Th}$ and $A_K$ for the syenogranite samples but a negative distribution for the alkali feldspar granite samples. In the quartz syenite samples, $A_{Th}$ has a negative asymmetric distribution and $A_K$ has a positive asymmetric distribution. Moreover, the kurtosis values illustrate the distribution probability of Preakness in $A_K$ for alkali feldspar granite samples, while flatness was reported in the other $A_U$, $A_{Th}$, and $A_K$ values, where the values are negative. The higher concentration of $^{238}$U, $^{232}$Th and $^{40}$K are associated with the presence of radioactive minerals concentrated inside the granitoids faults [51,52]. Furthermore, hydrothermal mechanisms, particularly alkaline fluids, are accountable for the percolation of meteroic water, which causes uranium and thorium minerals (uranophane, autunite, kasolite and curite) to remobilize and become stuck in granitoid joints, cracks and fractures [30,31].

4.6.2. $D_{air}$ and AED

$D_{air}$ is the radiological factor employed to assess exposure to gamma radiation from terrestrial radioactivity above 1 m from the surface of the ground. The following Equation (1) was utilized to compute the absorbed dose rate [53]:

$$D_{air} \text{ (nGy/h)} = 0.430 \, (A_U) + 0.666 \, (A_{Th}) + 0.042 \, (A_K) \tag{1}$$

$A_U$, $A_{Th}$ and $A_K$ refer to $^{238}$U, $^{232}$Th and $^{40}$K activity concentrations, respectively. The computed data of $D_{air}$ are listed in Table 3. The results of $D_{air}$ show the average values of $D_{air}$ are identified in the granitoids samples as follows: $D_{air}$ (syenogranite = 217 nGy/h) > $D_{air}$ (quartz syenite = 151 nGy/h) > $D_{air}$ (alkali feldspar granite = 111 nGy/h), which is higher than the reported limit (59 nGy/h) [29,54]. Statistically, the maximum values measured for $D_{air}$ are 218, 138 and 173 nGy/h for syenogranite, alkali feldspar granite and quartz syenite, respectively, while the minimum values are 151, 82 and 137 nGy/h, for syenogranite, alkali feldspar granite and quartz syenite, respectively. This displays that the granitoids of the studied area are not appropriate for use in several construction applications. Public exposure to gamma radiation released from granitoids can be assessed by the annual effective dose (AED), with two scenarios: outdoor (AED$_{out}$ with occupancy factor of 0.2) and indoor (AED$_{in}$ with occupancy factor of 0.8), where the exposure time is 8760 h. AED values are estimated according to the following Equations (2) and (3) [55,56]:

$$\text{AED}_{out} \text{ (mSv/y)} = D_{air} \times 0.2 \times 8760 \, \text{h} \times 0.7 \, \text{Sv/Gy} \times 10^{-6} \tag{2}$$

$$\text{AED}_{in} \text{ (mSv/y)} = D_{air} \times 0.8 \times 8760 \, \text{h} \times 0.7 \, \text{Sv/Gy} \times 10^{-6} \tag{3}$$

The dose conversion factor (DCF = 0.7 Sv/Gy) was applied in the Equations to illustrate the effect of gamma radiation on the public. Table 3 displays AED$_{out}$ and AED$_{in}$ values are highest in syenogranite and lowest in alkali feldspar granite. The average values of AED$_{out}$ (0.27, 0.14 and 0.19 mSv/y for syenogranite, alkali feldspar granite and quartz syenite,

respectively) and $AED_{in}$ (1.06, 0.54 and 0.19 mSv/y for syenogranite, alkali feldspar granite and quartz syenite, respectively) exceed the recommended UNSCEAR limit (0.07 mSv/y—outdoor and 0.41 mSv/y—indoor) [29]. The distribution of heavy radioactive minerals through the granitoids leads to high doses of gamma radiation. Thus, the application of various granitoids in building construction leads to different adverse effects for the public, such as cancer, coronary heart disease, deoxyribonucleic acid (DNA) damage and tissue degeneration [57].

### 4.6.3. AGDE and ELCR

Applied radiological parameters include the annual gonadal dose equivalent (AGDE, mSv/y), utilized to assess the effects of gamma radiation on the gonads. AGDE is computed by the following Equation (4) [58]:

$$AGDE \ (mSv/y) = 3.09 \times A_U + 4.18 \times A_{Th} + 0.314 \times A_K \tag{4}$$

AGDE was determined for all granitoids samples (Table 3). Average AGDE values ranged from 0.79 (alkali feldspar granite) to 1.53 (syenogranite) mSv/y. The average values are much higher than the reported limit of 0.3 mSv/y [29]. Thus, the granitoids in the studied area are not appropriate for building materials.

The toxic effects would most likely be realized if the public is exposed for a long time to the $\gamma$ radiation from granitoids, both outdoors and indoors. Thus, the excess lifetime cancer risk (ELCR) is estimated according to $AED_{out}$ and $AED_{in}$ through the life duration (PL = 70 years) and the factor of cancer risk (RF = 0.05 $Sv^{-1}$), as described by the International Commission of Radiation Protection (ICRP) [59]. The following Formulae (5, 6 and 7) are applied to calculate ELCR:

$$ELCR_{out} = AED_{out} \times PL \times RF \times 1000 \tag{5}$$

$$ELCR_{in} = AED_{in} \times PL \times RF \times 1000 \tag{6}$$

$$ELCR_t = ELCR_{out} + ELCR_{in} \tag{7}$$

The $ELCR_t$ values of the investigated granitoids are higher than the permissible value (0.00029) [59]. Figure 9 illustrates the range of average values, ranging from 2.38 (alkali feldspar granite) to 4.65 (syenogranite). Table 2 reveals the values of $ELCR_{out}$ in the granitoids are lower than $ELCR_{in}$. Therefore, the public cancer risk can be predicted based on years of life. It is strongly recommended to avoid using the granitoids in buildings and infrastructure.

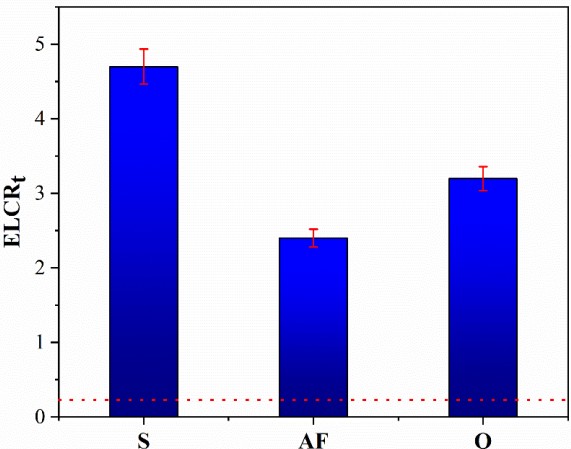

**Figure 9.** The average variation of total excess lifetime cancer risk (ELCRt) with granitoids type: syenogranite (S), alkali feldspar granite (AF), and quartz syenite (Q), the red line means the permissible limit.

### 4.6.4. Effective Dose ($D_o$) to Various Body Organs

The effective dosage rate provided to a certain organ is computed using Equation (8) below [60]; the computation of $D_o$ depends on the scenario of exposure (outdoor—$AED_{out}$ and AEDin—indoor). The differentiation of effective dose between organs is governed by the dose conversion factor (F), dictated by the ICRP for each organ (0.46—liver, 0.58—ovaries, 0.62—kidneys, 0.64—lungs, 0.69—bone marrow, 0.82—testes and 0.68—whole body) [61].

$$D_o \ (\text{mSvy}^{-1}) = \text{AED} \times \text{F} \qquad (8)$$

Figure 10 shows the $D_o$ for different organs, where $D_o$ indicates how much radiation is stored in various human tissues and organs after a year of exposure. The highest $D_o$ values are detected through the indoor exposure to gamma radiation for all studied organs, where it ranged from 0.25 mSv/y (alkali feldspar granite) to 0.86 mSv/y (syenogranite). Moreover, the lowest $D_o$ values are predicated with outdoor exposure to gamma radiation, where it varied from 0.06 mSv/y (alkali feldspar granite) to 0.21 mSv/y (syenogranite). The estimated doses to the numerous organs tested are all less than the permissible worldwide threshold dose of 1.0 mSv per year, according to these data. The testes receive a higher dosage than the other tissues, and the liver receives a lesser dose, which is explained by the rate of absorption of nutrients from diet [62]. This reveals that $\gamma$ radiation from granitoids in the examined area would not a considerable impact on the radiation dose reaching these adult organs.

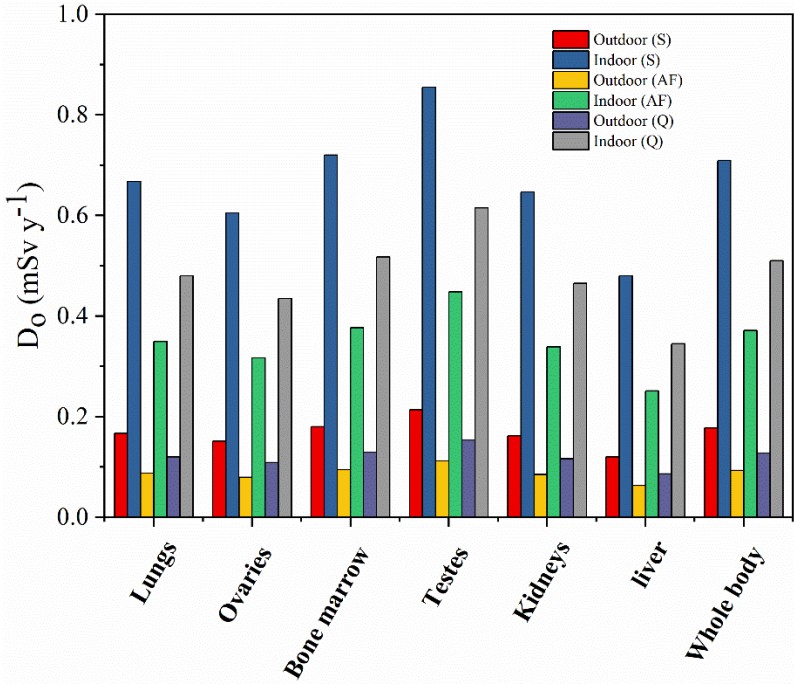

**Figure 10.** The effective dosage rate $D_o$ (mSv/y) associate with various tissues from air dose outdoor and indoor of syenogranite, alkali feldspar granite, and quartz syenite.

### 4.7. Statistical Analysis

Statistical analysis was performed to illustrate correlations between the radionuclide concentrations and the mentioned radiological hazards associated with the granitoids. Among statistical analyses, Pearson correlation matrix (PCM) and cluster analysis (CA) are suggested. Tables 5–7 indicate the PCM as follow: 1—for syenogranite, the radiological hazards are contributed to by heavy radioactive minerals. 2—for alkali feldspar granite, however, there is a weak correlation ($R^2 = 0.39$) between $^{238}$U and $^{232}$Th; uranium, thorium and their minerals are the main contributors to the radiological hazards. 3—for quartz syenite, the main contributor of radioactive effects is thorium and its minerals. It can be

seen in Tables 5–7 that the AGDE has very strong correlations ($R^2$ = 0.94 and 0.92) with $^{238}$U for alkali feldspar granite and quartz syenite. In addition, there is a very strong correlation ($R^2$ = 0.96) with $^{232}$Th for syenogranite. This also establishes the geological properties of the granitoids in the examined zone, where weathering has created heavy radioactive minerals, including uranothorite and thorite in addition to zircon, monazite, bastnäsite, synchysite, xenotime and allanite.

**Table 5.** The Pearson correlation matrix (PCM) of $^{238}$U, $^{232}$Th, $^{40}$K activity concentrations and the radiological variables of syenogranite.

| | $^{238}$U | $^{232}$Th | $^{40}$K | $D_{air}$ | $AED_{out}$ | $AED_{in}$ | AGDE | $ELCR_{out}$ | $ELCR_{in}$ | $ELCR_t$ |
|---|---|---|---|---|---|---|---|---|---|---|
| $^{238}$U | 1 | | | | | | | | | |
| $^{232}$Th | 0.40 | 1 | | | | | | | | |
| $^{40}$K | 0.58 | 0.67 | 1 | | | | | | | |
| $D_{air}$ | 0.82 | 0.83 | 0.83 | 1 | | | | | | |
| $AED_{out}$ | 0.82 | 0.83 | 0.83 | 1 | 1 | | | | | |
| $AED_{in}$ | 0.82 | 0.83 | 0.83 | 1 | 1 | 1 | | | | |
| AGDE | 0.62 | 0.96 | 0.74 | 0.94 | 0.94 | 0.94 | 1 | | | |
| $ELCR_{out}$ | 0.82 | 0.83 | 0.83 | 1 | 1 | 1 | 0.94 | 1 | | |
| $ELCR_{in}$ | 0.82 | 0.83 | 0.83 | 1 | 1 | 1 | 0.94 | 1 | 1 | |
| $ELCR_t$ | 0.82 | 0.83 | 0.83 | 1 | 1 | 1 | 0.94 | 1 | 1 | 1 |

**Table 6.** The Pearson correlation matrix (PCM) of $^{238}$U, $^{232}$Th, $^{40}$K activity concentrations and the radiological variables of alkali feldspar granite.

| | $^{238}$U | $^{232}$Th | $^{40}$K | $D_{air}$ | $AED_{out}$ | $AED_{in}$ | AGDE | $ELCR_{out}$ | $ELCR_{in}$ | $ELCR_t$ |
|---|---|---|---|---|---|---|---|---|---|---|
| $^{238}$U | 1 | | | | | | | | | |
| $^{232}$Th | 0.39 | 1 | | | | | | | | |
| $^{40}$K | 0.23 | 0.19 | 1 | | | | | | | |
| $D_{air}$ | 0.72 | 0.80 | 0.61 | 1 | | | | | | |
| $AED_{out}$ | 0.72 | 0.80 | 0.61 | 1 | 1 | | | | | |
| $AED_{in}$ | 0.72 | 0.80 | 0.61 | 1 | 1 | 1 | | | | |
| AGDE | 0.94 | 0.56 | 0.48 | 0.90 | 0.90 | 0.90 | 1 | | | |
| $ELCR_{out}$ | 0.72 | 0.80 | 0.61 | 1 | 1 | 1 | 0.90 | 1 | | |
| $ELCR_{in}$ | 0.72 | 0.80 | 0.61 | 1 | 1 | 1 | 0.90 | 1 | 1 | |
| $ELCR_t$ | 0.72 | 0.80 | 0.61 | 1 | 1 | 1 | 0.90 | 1 | 1 | 1 |

The correlation among radiological factors is studied by clustering analysis (HCA) and presented in Figure 11. The three clusters are planned in the dendrogram of the examined results of the granitoids. Figure 11a illustrates the clusters for syenogranite are: Cluster I—$^{232}$Th and AGDE; and Cluster II—$^{238}$U and the rest of the radiological hazards). Furthermore, Figure 11b,c reveals that the clusters for alkali feldspar granite and quartz syenite are: Cluster I—$^{238}$U and AGDE; and Cluster II—$^{232}$Th and the rest of the radiological hazards. CA showed the radioactivity of granitoids is related to the concentration of $^{238}$U and $^{232}$Th, which agrees with the PCM.

**Table 7.** The Pearson correlation matrix (PCM) of $^{238}$U, $^{232}$Th, $^{40}$K activity concentrations and the radiological variables of quartz syenite.

| | $^{238}$U | $^{232}$Th | $^{40}$K | $D_{air}$ | $AED_{out}$ | $AED_{in}$ | AGDE | $ELCR_{out}$ | $ELCR_{in}$ | $ELCR_t$ |
|---|---|---|---|---|---|---|---|---|---|---|
| $^{238}$U | 1 | | | | | | | | | |
| $^{232}$Th | 0.39 | 1 | | | | | | | | |
| $^{40}$K | −0.69 | −0.12 | 1 | | | | | | | |
| $D_{air}$ | 0.38 | 0.81 | 0.27 | 1 | | | | | | |
| $AED_{out}$ | 0.38 | 0.81 | 0.27 | 1 | 1 | | | | | |
| $AED_{in}$ | 0.38 | 0.81 | 0.27 | 1 | 1 | 1 | | | | |
| AGDE | 0.92 | 0.58 | −0.37 | 0.71 | 0.71 | 0.71 | 1 | | | |
| $ELCR_{out}$ | 0.38 | 0.81 | 0.27 | 1 | 1 | 1 | 0.71 | 1 | | |
| $ELCR_{in}$ | 0.38 | 0.81 | 0.27 | 1 | 1 | 1 | 0.71 | 1 | 1 | |
| $ELCR_t$ | 0.38 | 0.81 | 0.27 | 1 | 1 | 1 | 0.71 | 1 | 1 | 1 |

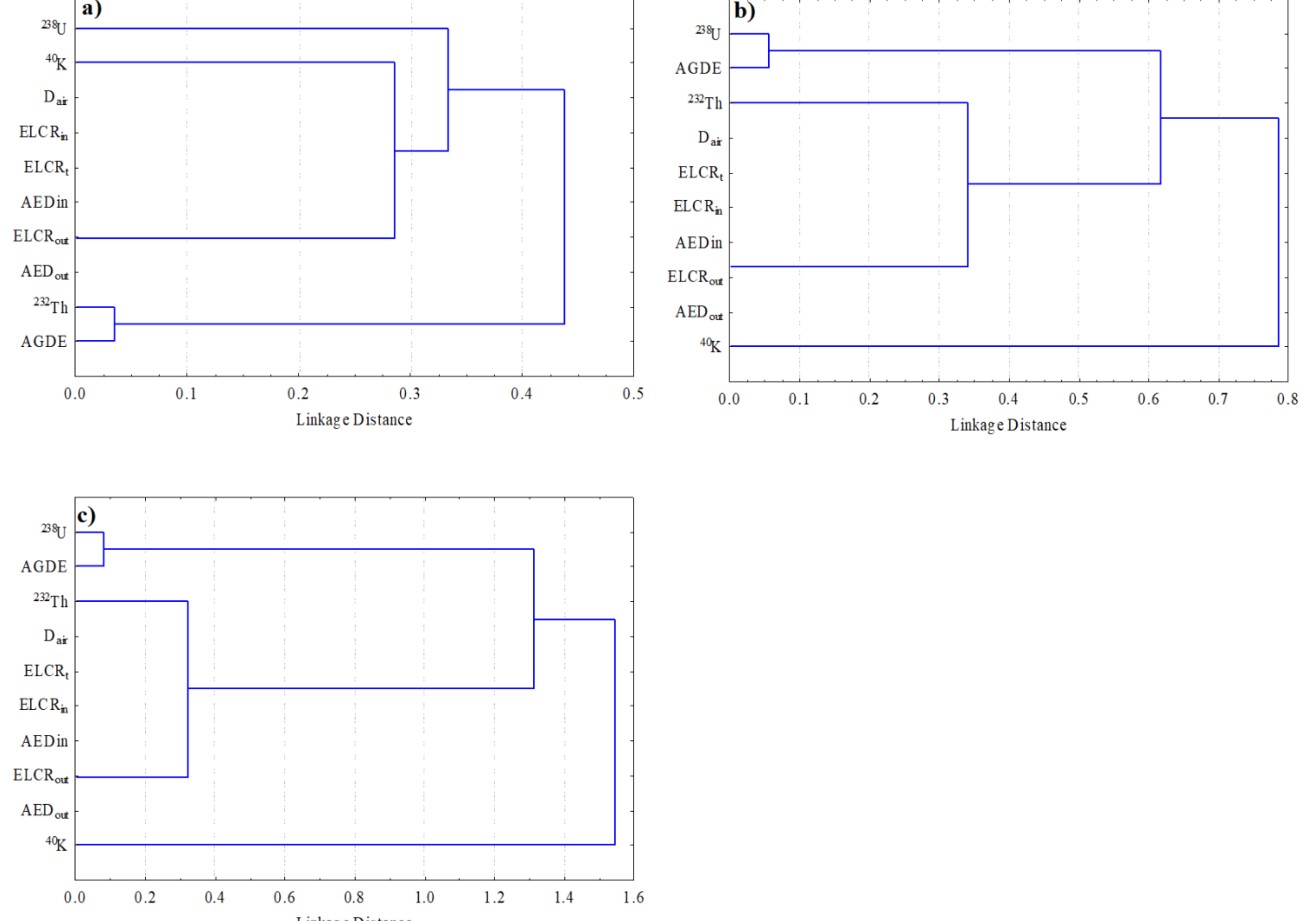

**Figure 11.** Linkage between various statistical radiological variables among the studied granitoids; (**a**) syenogranite, (**b**) alkali feldspar granite, (**c**) quartz syenite.

## 5. Conclusions

Radioelement measurements show that syenogranite and quartz syenite as well as microgranite dikes possess higher concentrations (of K, eU and eTh and their ratios) than alkali feldspar granite. A linear increase of the eTh–eU ratio is observed within larger Nikeiba granitoids plutons. Most of the samples are grouped around an eTh/eU ratio of

2.7 to 5.5 during magmatic fractionation. The higher radioelement measurements related to hydrothermal alteration (albitization, episyenitization and hematitization). In addition, accessory minerals—uranothorite, thorite, zircon, monazite, bastnäsite, synchysite, xenotime and allanite—are associated with syenogranite and quartz syenite. Geochemically, syenogranite, alkali feldspar granite and quartz syenite are enriched in large-ion lithophile elements (LILE; Ba, Rb, Sr) high field-strength elements (HFSE; Y, Zr and Nb) but have decreased Ce, reflecting their alkaline affinity. They exhibit calc–alkaline affinity and metaluminous characteristics and were emplaced in within-plate conditions under an extensional anorogenic environment. Geochemically, U and Th concentrations increase with differentiation, especially in syenogranite and quartz syenite, whereas alkali feldspar granite shows a decrease in their contents. The activity concentrations of $A_U$, $A_{Th}$ and $A_K$ are extremely higher than the average of worldwide value. Furthermore, radiological hazard variables such as AGDE and ELCR were calculated for the studied samples and shown to be higher than the recommended limits. Statistical analysis illustrates that high doses are contributed to by the presence of radioactive minerals in the granitoids. Thus, the granitoids are unsuitable to be employed in construction.

**Author Contributions:** Conceptualization, A.E.A.G., K.G.A. and M.Y.H.; methodology, A.E.A.G. and H.E.; software, A.E.A.G. and M.Y.H.; validation, K.A. and M.S.S.; formal analysis, A.E.A.G. and M.Y.H.; investigation, H.E. and M.M.; resources, A.E.A.G. and M.S.S.; data curation, K.G.A. and H.E.; writing—original draft preparation, A.E.A.G., M.M. and M.Y.H.; writing—review and editing, A.E.A.G. and M.Y.H.; visualization, A.E.A.G., M.S.S. and M.Y.H.; supervision, M.S.S. and H.E.; project administration, A.E.A.G. and K.G.A.; funding acquisition, K.A. All authors have read and agreed to the published version of the manuscript.

**Funding:** This research received no external funding.

**Data Availability Statement:** Not applicable.

**Conflicts of Interest:** The authors declare no conflict of interest.

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
