# Peer review of "Cancer Risk Assessment and Geochemical Features of Granitoids at Nikeiba, Southeastern Desert, Egypt"

_minerals, doi:10.3390/min12050621_

Round 1

Reviewer 1 Report

General comment

This study is also significant from a public health point of view, as it examines the geology in Nikeiba, South Eastern Desert, Egypt, and reveals that some of the materials are inappropriate for use as building materials especially for children.

Specific comments

Line99

The definition of “eU” is clear for readers as “equivalent uranium”? The same applies below.

Line 102

It would be good to put a space between the unit and the number. The same applies below.

Line 111

“T.C, in (Ur) which defined as unit of radioelement concentration”

Does it mean that “unit of radioelement concentration” is Bq/kg?

The explanation of the unit “Ur” would be helpful for Non-specialized readers.

Line 173

“The eU/eTh versus eTh diagram (Figure 2c) shows a reverse relationship”

Is it supported by statistical analysis?

Line 407

Equation (4)

The dimension of AGDE is dose per year.

The unit of AGDE in the reference 57 is µSv/y. So that the unit is different from the formula shown in the reference, is not that a problem?

Line 419

Equation (5)

The dimension of ELCR is dimensionless.

The multiplication sign should be corrected.

Line 438

Equation (8)

“mSvy-1

The units of the paper are not consistently described in the paper such as “mSvy-1”, “mSv/y” and “mSv y-1”.

Reference

  1.  

The literature is not presented.

Table 1

K%

Is “K%” the compositional percentage of potassium as described at line 81?

If so, “Mass percent concentration of K might be more understandable, even though it would be obvious for all readers. The same applies below.

Table 2

CIPW Norm

I assume it is self-explanatory for any readers of this paper, but it might be helpful to have either a reference or an explanation of the concept.

Table 3 on page 19

The unit for AED would be mSv/y not mSv. The same applies below.

Table 3 on page 21 should be Table 4

Is it reasonable to have a higher value for the geometric mean than for the arithmetic mean?

Figure 2

The arrows in the diagram are somewhat arbitrary and should be removed. If authors really want to keep them, why not add an explanation to the legend in the figure?

Figure 9

The values on the vertical axis are multiplied by 1,000.

The definition of error bar is needed even though it would be obvious for all readers.

It may be better to avoid using abbreviations in the titles of figures and tables. The same applies below.

Author Response

Dear Reviewer,

Please find attached the submission of the carefully revised version of the manuscript in Ref., following the minor comments and modification of the Reviewer.

Below is a detailed list of the changes made in response to the Reviewer’s minor comments which outlines every change made a point by point. The changes are marked in the manuscript text (track change).

Specific comments

Line99

The definition of “eU” is clear for readers as “equivalent uranium”? The same applies below.

Response: it explained in the text

Line 102

It would be good to put a space between the unit and the number. The same applies below.

 Response: it is corrected in the manuscript.

Line 111

“T.C, in (Ur) which defined as unit of radioelement concentration”

Does it mean that “unit of radioelement concentration” is Bq/kg?

The explanation of the unit “Ur” would be helpful for Non-specialized readers.

Response:

1Ur ≈ 1 eU(ppm) = 12.27 Bq/kg (International Atomic Energy Agency (IAEA, 1976)

International Atomic Energy Agency "IAEA", 1976: Radiometric Reporting Methods and calibration in uranium Exploration, Technical Reports series No.174, IAEA, Vienna.

  Line 173

“The eU/eTh versus eTh diagram (Figure 2c) shows a reverse relationship”

Is it supported by statistical analysis?

Response:

The eU/eTh ratio increases with eTh (Figure 2c) for most plotted alkali feldspar granite, syenogranite and quartz syenite samples. The

Line 407

Equation (4)

The dimension of AGDE is dose per year.

The unit of AGDE in the reference 57 is µSv/y. So that the unit is different from the formula shown in the reference, is not that a problem?

 Response: It is not problem. The unit corrected in the equation.

Line 419

Equation (5)

The dimension of ELCR is dimensionless.

 Response: it is corrected

The multiplication sign should be corrected.

  Response: it is corrected

Line 438

Equation (8)

“mSvy-1

The units of the paper are not consistently described in the paper such as “mSvy-1”, “mSv/y” and “mSv y-1”.

  Response: corrected in the manuscript mSv/y

Reference

  1.  

The literature is not presented.

Response: it presented in the section “4.6.4. Effective dose (Do) to various body organs”.

Table 1

K%

Is “K%” the compositional percentage of potassium as described at line 81?

If so, “Mass percent concentration of K might be more understandable, even though it would be obvious for all readers. The same applies below.

Response: The radioelement measurements were taken using a single measurement in each site for eTh (ppm) and eU (ppm) and K (%) at each measured station.

 Table 2

CIPW Norm

I assume it is self-explanatory for any readers of this paper, but it might be helpful to have either a reference or an explanation of the concept.

Response: The CIPW norm proposed one hundred years ago is still a useful scheme because abundances of normative minerals are required for a proper rock classification such as that recommended by the IUGS.

The CIPW Norm (Cross, Iddings, Pirrson and Washington) is based upon assumptions about the order of mineral formation and known phase relationships of rocks and minerals, using simplified mineral formulas (Johannsen, 1931; Kelsey, 1965; Hutchison, 1974; Cox et al., 1979; Le Maitre, 1982; Ragland, 1989; Rollinson, 1993; Verma et al., 2003; González-Guzmán 2016).

Table 3 on page 19

The unit for AED would be mSv/y not mSv. The same applies below.

Response: it is corrected in the manuscript.

Table 3 on page 21 should be Table 4

Response: it is corrected

Is it reasonable to have a higher value for the geometric mean than for the arithmetic mean?

Response: the difference between the two values is due to the statistical calculations.

 Figure 2

The arrows in the diagram are somewhat arbitrary and should be removed. If authors really want to keep them, why not add an explanation to the legend in the figure?

Response: Have done (arbitrary removed)

Figure 9

The values on the vertical axis are multiplied by 1,000.

Response: “1,000” added in the equations 5 and 6.

The definition of error bar is needed even though it would be obvious for all readers.

Response: the errors bars display the error percentage in the results.

It may be better to avoid using abbreviations in the titles of figures and tables. The same applies below.

Response: it is corrected

We thank the Reviewers a lot for the useful and valuable comments that have helped improve the manuscript.

Hoping that all the careful review is sufficient for the direct acceptance of the manuscript, thank you for your time and consideration.

Best wishes,

Reviewer 2 Report

  1. Introduction: What are the scientific problems, and why should research be carried out.
  2. Geologic setting: Please supplement the main intrusive mineral content.
  3. Line 40, 44: “Central Eastern Desert” should be replaced with “Central East Desert”, throughout the paper.

Author Response

Dear Reviewer,

Please find attached the submission of the carefully revised version of the manuscript in Ref., following the minor comments and modification of the Reviewer.

Below is a detailed list of the changes made in response to the Reviewer’s minor comments which outlines every change made a point by point. The changes are marked in the manuscript text (track change).

Introduction: What are the scientific problems, and why should research be carried out.

Response: it is improved and described in the manuscript.

Geologic setting: Please supplement the main intrusive mineral content.

Response: Syenogranite is medium to coarse grained, whitish to pale pink, buff, reddish brown colors, jointed, strongly weathered and exfoliated. They contain xenoliths up to 1m of subangular metatuffs along their outer periphery. It is composed mainly of K-feldspar, quartz, plagioclase, and biotite. Zircon, allanite, titanite, apatite, fluorite, and iron oxides are the main accessories, while chlorite and epidote are the main alteration products.  Alkali feldspar granite is coarse-grained, whitish, yellowish to pale white colors, highly weathered and holocrystalline equigranular rock. Porphyritic and micrographic textures are observed. It is essentially composed of K-feldspar, quartz, plagioclase, and biotite. Zircon, apatite and iron opaque minerals are accessories, whereas chlorite, and seicite are alteration products. Quartz syenite is medium to coarse grained, dark grey to pale greenish grey, pale pink in colors and moderate to high relief. It is highly weathered, exfoliated, holocrystalline, hypidiomorphic granular rock and microscopically composed of K-feldspars, quartz, biotite, riebeckite, arfvedsonite and very subordinate amount of plagioclase. Zircon, apatite, allanite, iron oxides and opaque minerals are accessories, whereas muscovite, chlorite, sericite, epidote, and carbonates are the main alteration products.

Line 40, 44: “Central Eastern Desert” should be replaced with “Central East Desert”, throughout the paper.

Response: the authors believe “Central Eastern Desert” is correct and used in other publications.

We thank the Reviewers a lot for the useful and valuable comments that have helped improve the manuscript.

Hoping that all the careful review is sufficient for the direct acceptance of the manuscript, thank you for your time and consideration.

Best wishes,